# PREFERENCE ELICITATION FOR OFFLINE REINFORCEMENT LEARNING

**Alizée Pace**
ETH AI Center, ETH Zürich
MPI for Intelligent Systems, Tübingen
alizee.pace@ai.ethz.ch

**Bernhard Schölkopf**
MPI for Intelligent Systems &
ELLIS Institute Tübingen

**Gunnar Rätsch**
ETH Zürich

**Giorgia Ramponi**
University of Zürich

## ABSTRACT

Applying reinforcement learning (RL) to real-world problems is often made challenging by the inability to interact with the environment and the difficulty of designing reward functions. Offline RL addresses the first challenge by considering access to an offline dataset of environment interactions labeled by the reward function. In contrast, Preference-based RL does not assume access to the reward function and learns it from preferences, but typically requires an online interaction with the environment. We bridge the gap between these frameworks by exploring efficient methods for acquiring preference feedback in a fully offline setup. We propose Sim-OPRL, an offline preference-based reinforcement learning algorithm, which leverages a learned environment model to elicit preference feedback on simulated rollouts. Drawing on insights from both the offline RL and the preference-based RL literature, our algorithm employs a pessimistic approach for out-of-distribution data, and an optimistic approach for acquiring informative preferences about the optimal policy. We provide theoretical guarantees regarding the sample complexity of our approach, dependent on how well the offline data covers the optimal policy. Finally, we demonstrate the empirical performance of Sim-OPRL in various environments.

## 1 INTRODUCTION

While reinforcement learning (RL) (Sutton and Barto, 2018) achieves excellent performance in various decision-making tasks (Kendall et al., 2019; Mirhoseini et al., 2020; Degrave et al., 2022), its practical deployment remains limited by the requirement of direct interaction with the environment. This can be impractical or unsafe in real-world scenarios. For example, patient management and treatment in intensive care units involve complex decision-making that has often been framed as a reinforcement learning problem (Raghu et al., 2017; Komorowski et al., 2018). However, the timing, dosage, and combination of treatments required are critical to patient safety, and incorrect decisions can lead to severe complications or death, making the use of traditional RL algorithms unfeasible (Gottesman et al., 2019; Tang and Wiens, 2021). Offline RL emerges as a promising solution, allowing policy learning from entirely observational data (Levine et al., 2020).

Still, a challenge with Offline RL is its requirement for an explicit reward function. Quantifying the numerical value of taking a certain action in a given environment state is challenging in many applications (Yu et al., 2021). Preference-based RL offers a compelling alternative, relying on comparisons between different actions or trajectories (Wirth et al., 2017) and being often easier for humans to provide (Christiano et al., 2017). In medical settings, for instance, clinicians may be queried for feedback on which trajectories lead to favorable outcomes. Unfortunately, most algorithms for preference acquisition require environment interaction (Saha et al., 2023; Chen et al., 2022; Lindner et al., 2021) and are therefore not applicable to the offline setting.

Lack of environment interaction and reward learning are thus two critical challenges for real-world RL deployment that are rarely tackled jointly. In this work, we address the problem of prefer-

ence elicitation for offline reinforcement learning by asking: *What trajectories should we sample to minimize the number of human queries required to learn the best offline policy?* This presents a challenging problem as it combines learning from offline data and active feedback acquisition, two frameworks that require opposing inductive biases for conservatism and exploration, respectively.

To the best of our knowledge, the only strategy proposed in prior work is to acquire feedback directly over samples within an offline dataset of trajectories (Shin et al., 2022, Offline Preference-based Reward Learning (OPRL)). We propose an alternative solution that queries feedback on *simulated rollouts* by leveraging a learned environment model. Our offline preference-based reinforcement learning algorithm, Sim-OPRL, strikes a balance between conservatism and exploration by combining pessimism when handling states out-of-distribution from the observational data (Jin et al., 2021; Zhan et al., 2023a), and optimism in acquiring informative preferences about the optimal policy (Saha et al., 2023; Chen et al., 2022). We study the efficiency of our approach through both theoretical and empirical analysis, demonstrating the superior performance of Sim-OPRL across various environments.

Our contributions are the following: (1) In Section 3, we first formalize the new problem setting of preference elicitation for offline reinforcement learning, which allows for **complementing offline data with preference feedback**. This framework is crucial for real-world RL applications where direct environment interaction is unsafe or impractical and reward functions are challenging to design manually, yet experts can be queried for their knowledge. (2) In Section 5, we provide theoretical guarantees on eliciting preference feedback over samples from an offline dataset, complementing work of Shin et al. (2022). (3) Next, in Section 6, we propose our own **efficient preference elicitation algorithm** based on simulated rollouts in a learned environment model, and establish its improved theoretical guarantees. (4) Finally, we develop a practical implementation of our algorithm and demonstrate its **empirical efficiency** and scalability across various decision-making environments.

## 2 RELATED WORK

Our problem setting shares similarities with Offline RL and Preference-based RL, which we summarize below. We position ourselves relative to our closest related works in Table 1 and extend our discussion in Appendix B.

**Offline RL.** Offline Reinforcement Learning has gained significant traction in recent years, as the practicality of training RL agents without environment interaction makes it relevant to real-world applications (Levine et al., 2020). However, learning from observational data only is a source of bias in the model, as the data may not cover the entire state-action space. Offline RL algorithms therefore output pessimistic policies, which has been shown to minimize suboptimality (Jin et al., 2021). Model-based approaches show particular promise for their sample efficiency (Yu et al., 2020; Kidambi et al., 2020; Rigter et al., 2022; Zhai et al., 2024; Uehara and Sun, 2021). In this work, we study the setting where reward signals are unavailable and must be estimated by actively querying preference feedback.

**Preference-based RL.** Rather than accessing numerical reward values for each state-action pair as in traditional online RL, preference-based RL learns the reward model through collecting pairwise preferences over trajectories (Wirth et al., 2017). Different preference elicitation strategies have been proposed for this framework, generally based on knowing the transition model exactly or on having access to the environment for rollouts (Christiano et al., 2017; Saha et al., 2023; Chen et al., 2022; Lindner et al., 2021; Zhan et al., 2023b; Sadigh et al., 2018; Brown et al., 2020). Prior theoretical and empirical work (Lindner et al., 2021; Chen et al., 2022) show that, in this setting, the most efficient preference elicitation strategy is to actively reduce the set of candidate optimal

Table 1: **Comparison of related work on preference elicitation.**

| Framework | Offline | Policy-Based Sampling | Robustness Guarantees | Practical Implementation |
|---|---|---|---|---|
| PbOP (Chen et al., 2022) | ✗ | ✓ | ✓ | ✗ |
| MoP-RL (Liu et al., 2023) | ✗ | ✗ | ✗ | ✓ |
| REGIME (Zhan et al., 2023b) | ✗ | ✓ | ✓ | ✗ |
| FREEHAND (Zhan et al., 2023a) | ✓ | ✗ | ✓ | ✗ |
| OPRL (Shin et al., 2022) | ✓ | ✗ | ✗ | ✓ |
| **Sim-OPRL (Ours)** | ✓ | ✓ | ✓ | ✓ |

policies, rather maximize information gain on the reward function – our theoretical and empirical results reach the same conclusion for the offline setting.

**Offline Preference-based RL.** The development of preference-based RL algorithms based on offline data only is critical to settings where environment interaction is not feasible for safety and efficiency reasons. Still, this framework remains largely unexplored in the literature. While Zhu et al. (2023); Zhan et al. (2023a) demonstrate the value of pessimism in offline preference-based reinforcement learning, they do not consider how to query feedback actively. On the other hand, Shin et al. (2022) propose an empirical comparison of different preference sampling trajectories from an offline trajectories buffer. In Section 5, we provide a theoretical analysis of their approach, then propose an alternative sampling strategy based on simulated trajectory rollouts in Section 6, which benefits from both theoretical motivation and superior empirical performance.

# 3 PROBLEM FORMULATION

## 3.1 PRELIMINARIES

**Markov Decision Process.** We consider the episodic Markov Decision Process (MDP), defined by the tuple $\mathcal{M} = (\mathcal{S}, \mathcal{A}, H, T, R)$, where $\mathcal{S}$ is the state space, $\mathcal{A}$ is the action space, $H$ is the episode length, $T : \mathcal{S} \times \mathcal{A} \to \Delta_{\mathcal{S}}$ is the transition function, $R : \mathcal{S} \times \mathcal{A} \to \mathbb{R}$ is the reward function. We assume an initial state $s_0$, but our analysis generalizes to a fixed initial state distribution. At time $t$, the environment is at state $s_t \in \mathcal{S}$ and an agent selects an action $a_t \in \mathcal{A}$. The agent then receives a reward $R(s_t, a_t)$ and the environment transitions to state $s_{t+1} \sim T(\cdot|s_t, a_t)$. We describe an agent's behavior through a policy function $\pi : \mathcal{S} \to \Delta_{\mathcal{A}}$, such that $\pi(a|s)$ is the probability of taking action $a$ in state $s$. Let $\tau = (s_0, a_0, \dots s_H, a_H)$ denote the trajectory of state-action pairs of an interaction episode with the environment. With an abuse of notation, we also write $R(\tau) = \sum_t R(s_t, a_t)$. Let $d_T^\pi$ denote the distribution of trajectories (or state-action pairs, overloading notation) induced by rolling out policy $\pi$ in transition model $T$. We denote the expected return of policy $\pi$ as $V_{T,R}^\pi = \mathbb{E}_{\tau \sim d_T^\pi}[R(\tau)]$, and $\pi^* = \arg\max_\pi V_{T,R}^\pi$ denotes the optimal policy in $\mathcal{M}$.

**Preference-based Reinforcement Learning.** Rather than observing rewards for each state and action, we receive preference feedback over trajectories. For a pair of trajectories $(\tau_1, \tau_2)$, we obtain binary feedback $o \in \{0, 1\}$ about whether $\tau_1$ is preferred to $\tau_2$. We define the preference function $P_R$ and assume that preference labels follow the Bradley-Terry model (Bradley and Terry, 1952):

$$P_R(\tau_1 \succ \tau_2) := P(o = 1|\tau_1, \tau_2) = \frac{\exp(R(\tau_1))}{\exp(R(\tau_1)) + \exp(R(\tau_2))} = \sigma(R(\tau_1) - R(\tau_2)), \quad (1)$$

where $\succ$ denotes a preference relationship and $\sigma$ is a monotonous increasing link function. Within this framework, *preference elicitation* refers to the process of sampling preferences to obtain information about both the preference function and the system dynamics (Wirth et al., 2017).

## 3.2 OFFLINE PREFERENCE ELICITATION

We assume access to an observational dataset of trajectories $\mathcal{D}_{\text{offline}} = \{\tau : \tau \sim d_T^{\pi_\beta}\}$, where $\pi_\beta$ is an unknown behavioural policy in $\mathcal{M}$. As in Offline RL, we do *not* have access to the environment to observe transition dynamics or rewards under alternative action choices. We assume *not* to have access to the reward function, but we can query preference feedback from a human to generate a dataset of preferences $\mathcal{D}_{\text{pref}} = \{(\tau_1, \tau_2, o)\}$.

**Optimality Criterion.** Based only on our offline dataset $\mathcal{D}_{\text{offline}}$, our goal is to recover a policy $\hat{\pi}^*$ that minimizes suboptimality in the true environment with as few human preference queries as possible. Let $\pi^*_{\text{offline}}$ denote the *optimal offline policy* estimated based on the offline data, with access to the true reward function $R$, and let $\epsilon_T$ denote its suboptimality. Since preference elicitation only allows us to estimate the reward function, we do not aim to achieve a suboptimality less than $\epsilon_T$ – although this is not formally a lower bound for our problem, as shown in Appendix A.3. Our objective is then formalized as follows.

**Definition 3.1** (Optimality Criterion of Offline Preference Elicitation)**.** *Let $\pi^*$ be the optimal policy in $\mathcal{M}$ and $\hat{\pi}^*$ be the estimated optimal policy based on an offline dataset $\mathcal{D}_{\text{offline}}$ and $N_p > 0$ preference queries. Let $\epsilon_T$ be the inherent suboptimality assuming access to the true reward function.*

*We say that a sampling strategy is $(\epsilon, \delta, N_p)$-correct if for every $\epsilon \geq \epsilon_T$, with probability at least $(1 - \delta)$, it holds that $V_{T,R}^{\pi^*} - V_{T,R}^{\hat{\pi}^*} \leq \epsilon$.*

Our work is the first to formalize this important problem, which faces the challenge of balancing **exploration** when actively acquiring feedback and **bias mitigation** in learning from offline data.

**Function classes.** We estimate the reward function and transition kernel with general function approximation; let $\mathcal{F}_R$ and $\mathcal{F}_T$ denote the classes of functions considered respectively. We also assume a policy class $\Pi$. Our theoretical analysis also requires the following assumptions and definitions, which are standard in preference-based RL (Chen et al., 2022; Zhan et al., 2023a).

**Assumption 3.1** (Realizability). *The true reward function belongs to the reward class: $R \in \mathcal{F}_R$. The true transition function belongs to the transition class: $T \in \mathcal{F}_T$. The optimal policy belongs to the policy class: $\pi^* \in \Pi$.*

**Assumption 3.2** (Boundedness). *The reward function is bounded: $0 \leq \tilde{R}(\tau) \leq R_{\max}$ for all $\tilde{R} \in \mathcal{F}_R$ and all trajectories $\tau$.*

**Definition 3.2** ($\epsilon$-bracketing number). *Let $\mathcal{F}$ be a class of real functions $f : \mathcal{X} \to \mathbb{R}$. We say $(l, u)$ is an $\epsilon$-bracket if $l(x) \leq u(x)$ and $|u(x) - l(x)|_1 \leq \epsilon$ for all $x \in \mathcal{X}$. The $\epsilon$-bracketing number of $\mathcal{F}$, denoted $\mathcal{N}_{\mathcal{F}}(\epsilon)$, is the minimal number of $\epsilon$-brackets $(l^n, u^n)_{n=1}^N$ needed so that for any $f \in \mathcal{F}$, there is a bracket $i \in [N]$ containing it, meaning $l^i(x) \leq f(x) \leq u^i(x)$ for all $x \in \mathcal{X}$.*

Let $\mathcal{N}_{\mathcal{F}_R}(\epsilon)$ and $\mathcal{N}_{\mathcal{F}_T}(\epsilon)$ denote the $\epsilon$-bracketing numbers of $\mathcal{F}_R$ and $\mathcal{F}_T$ respectively. This measures the complexity of the function classes (Geer, 2000). For instance, with linear rewards $\mathcal{F}_R := \{R : \tau \to \theta^T \phi(\tau)\}$, the $\epsilon$-bracket number is bounded by $\log \mathcal{N}_{\mathcal{F}_R}(\epsilon) \leq \mathcal{O}(d \log \frac{BG}{\epsilon})$, where $\|\theta\|_2 \leq B$ and $\|\phi(\tau)\|_2 \leq G \, \forall \tau \in \mathcal{T}$, and $d$ is the dimension of the feature space (Zhan et al., 2023a).

**Definition 3.3** (Transition concentrability coefficient, Zhan et al. (2023a)). *The concentrability coefficient w.r.t. transition classes $\mathcal{F}_T$ and the optimal policy $\pi^*$ is defined as:*

$$C_T(\mathcal{F}_T, \pi^*) = \sup_{\tilde{T} \in \mathcal{F}_T} \left[ \frac{\mathbb{E}_{(s,a) \sim d_T^{\pi^*}}[|T(\cdot|s,a) - \tilde{T}(\cdot|s,a)|_1]}{\sqrt{\mathbb{E}_{(s,a) \sim \mathcal{D}_{\text{offline}}}[|T(\cdot|s,a) - \tilde{T}(\cdot|s,a)|_1^2]}} \right],$$

The concentrability coefficient measures the coverage of the optimal policy in the offline dataset. Note that $C_T$ is upper-bounded by the density-ratio coefficient: $C_T(\mathcal{F}_T, \pi^*) \leq \sup_{(s,a) \in \mathcal{S} \times \mathcal{A}} d_T^{\pi^*}(s,a)/d_T^{\pi_\beta}(s,a)$, where $\pi_\beta$ is the behavioural policy underlying $\mathcal{D}_{\text{offline}}$.

## 4 OFFLINE PREFERENCE-BASED RL WITH PREFERENCE ELICITATION

In this section, we propose a general framework for offline preference-based reinforcement learning. The next two sections propose two different preference elicitation strategies. As learning must be carried out in two distinct stages, with environment dynamics based on $\mathcal{D}_{\text{offline}}$ and reward learning on $\mathcal{D}_{\text{pref}}$, we adopt a model-based approach which we summarize in Algorithm 1.

**Model Learning.** We first leverage the offline data to learn a model of the environment dynamics, fitting a transition model $\hat{T}$ and an uncertainty function $u_T$ through maximum likelihood:

$$\hat{T} = \text{argmax}_{\tilde{T} \in \mathcal{F}_T} \mathbb{E}_{(s,a,s') \sim \mathcal{D}_{\text{offline}}} \left[ \log \tilde{T}(s'|s,a) \right],$$

$$u_T(s,a) = \max_{\tilde{T}_1, \tilde{T}_2 \in \mathcal{T}} |\tilde{T}_1(\cdot|s,a) - \tilde{T}_2(\cdot|s,a)|_1 \cdot R_{\max},$$

where $\mathcal{T} = \{\tilde{T} \in \mathcal{F}_T \mid \mathbb{E}_{(s,a,s') \sim \mathcal{D}_{\text{offline}}} \left[ \log \left( \hat{T}(s'|s,a)/\tilde{T}(s'|s,a) \right) \right] \leq \beta_T\}$, defining a confidence set over the maximum likelihood estimator (MLE), and $\beta_T$ is a margin hyperparameter.

**Iterative Preference Elicitation and Reward Learning.** As with the transition model, our algorithm estimates the reward function $\hat{R}$ and its uncertainty function through maximum likelihood over iteratively collected preference data $\mathcal{D}_{\text{pref}}$:

$$\hat{R} = \text{argmax}_{\tilde{R} \in \mathcal{F}_R} \mathbb{E}_{(\tau_1, \tau_2, o) \sim \mathcal{D}_{\text{pref}}} \left[ o \log P_{\tilde{R}}(\tau_1 \succ \tau_2) + (1 - o) \log P_{\tilde{R}}(\tau_2 \succ \tau_1) \right],$$

$$u_R(\tau) = \max_{\tilde{R}_1, \tilde{R}_2 \in \mathcal{R}} |\tilde{R}_1(\tau) - \tilde{R}_2(\tau)|_1,$$

---

**Algorithm 1** Offline Preference-based Reinforcement Learning with Preference Elicitation

---

**Input:** Observational trajectories dataset $\mathcal{D}_{\text{offline}}$. Significance $\delta \in (0, 1)$, preference budget $N_p$.
**Output:** $\hat{\pi}^*$
1: Estimate $\hat{T}$ and $u_T$ via maximum likelihood over the observational data $\mathcal{D}_{\text{offline}}$.
2: $\mathcal{D}_{\text{pref}} \leftarrow \emptyset$.
3: **for** $k = 1, ... N_p$ **do**
4:      Generate trajectory pairs $(\tau_1, \tau_2)$.          ▷ **Preference Elicitation**: Sections 5 and 6
5:      Collect preference label $o$ for $(\tau_1, \tau_2)$.
6:      $\mathcal{D}_{\text{pref}} \leftarrow \mathcal{D}_{\text{pref}} \cup \{(\tau_1, \tau_2, o)\}$.
7:      Estimate $\hat{R}$ and $u_R$ via maximum likelihood over the preference data $\mathcal{D}_{\text{pref}}$.
8: **end for**
9: $\hat{\pi}^* \leftarrow \text{argmax}_{\pi \in \Pi} \mathbb{E}_{\tau \sim d_{\hat{T}}^\pi}[\hat{R}(\tau) - u_R(\tau) - u_T(\tau)]$

---

where $\mathcal{R} = \{\tilde{R} \in \mathcal{F}_R \mid \mathbb{E}_{(\tau_1, \tau_2, o) \sim \mathcal{D}_{\text{pref}}} \left[\log \left(P_{\hat{R}}(\tau_1 \succ \tau_2)/P_{\tilde{R}}(\tau_1 \succ \tau_2)\right)\right] \leq \beta_R\}$ defines the confidence set and $\beta_R$ is a hyperparameter. We also define preference uncertainty between two trajectories $\tau_1, \tau_2$:

$$u_{P_R}(\tau_1, \tau_2) = \max_{\tilde{R}_1, \tilde{R}_2 \in \mathcal{R}} |P_{\tilde{R}_1}(\tau_1 \succ \tau_2) - P_{\tilde{R}_2}(\tau_1 \succ \tau_2)|_1. \tag{2}$$

The choice of trajectory sampling strategy for preference elicitation in line 4 (Algorithm 1) is critical to efficiently obtaining an $\epsilon$-optimal policy. We present two possible strategies in Sections 5 and 6.

**Pessimistic Policy Optimization.** Finally, our algorithm outputs a policy $\hat{\pi}^*$ that is optimal while ensuring robustness to modeling error. This means optimizing for the worst-case value function over the remaining transition and reward uncertainties (Levine et al., 2020):

$$\hat{\pi}^* = \text{argmax}_{\pi \in \Pi} \min_{\tilde{T} \in \mathcal{T}, \tilde{R} \in \mathcal{R}} V_{\tilde{T}, \tilde{R}}^\pi. \tag{3}$$

Based on this objective, we define the pessimistic transition and reward models as follows: $\hat{T}_{\text{inf}}, \hat{R}_{\text{inf}} = \text{argmin}_{\tilde{T} \in \mathcal{T}, \tilde{R} \in \mathcal{R}} \max_{\pi \in \Pi} V_{\tilde{T}, \tilde{R}}^\pi$. Our analysis provides a worst-case robustness guarantee when considering well-calibrated confidence intervals, as detailed in Sections 5.1 and 6.1. In other words, following prior work (Chen et al., 2022; Zhan et al., 2023a), our theoretical analysis assumes that modeling elements can be identified with no optimization error. We then complement this algorithmic framework with a flexible practical implementation in Section 6.3.

## 5    PREFERENCE ELICITATION FROM OFFLINE TRAJECTORIES

A first strategy for preference elicitation without environment interaction is to sample trajectories directly from the offline dataset. Shin et al. (2022) propose this approach as Offline Preference-based Reward Learning (OPRL), and design a uniform and uncertainty-sampling variant:

**OPRL Uniform Sampling**:                     $\tau_1, \tau_2 \sim \mathcal{D}_{\text{offline}}$

**OPRL Uncertainty Sampling**:     $\tau_1, \tau_2 = \text{argmax}_{\tau_1, \tau_2 \in \mathcal{D}_{\text{offline}}} u_{P_R}(\tau_1, \tau_2)$

We provide a theoretical analysis of the performance of OPRL.

### 5.1    THEORETICAL GUARANTEES.

We obtain the following result, proven in Appendix A.4. The suboptimality of the estimated policy $\hat{\pi}^*$ is bounded by the policy evaluation error for the optimal policy $\pi^*$. This error decomposes into a term depending on transition model estimation, and one on reward model estimation.

**Theorem 5.1.** *For any $\delta \in (0, 1]$, let $\beta_T = c_T^{\text{MLE}} \log(H\mathcal{N}_{\mathcal{F}_T}(1/N_o)/\delta)/N_o$ and $\beta_R = c_R^{\text{MLE}} \log(\mathcal{N}_{\mathcal{F}_R}(1/N_p)/\delta)/N_p$, where $N_o = H|\mathcal{D}_{\text{offline}}|$ is the number of observed transitions in the observational dataset and $c_T^{\text{MLE}}, c_R^{\text{MLE}}$ are universal constants. The policy $\hat{\pi}^*$ estimated by Algorithm 1, with preference elicitation based on offline trajectories, achieves the following suboptimal-*

*ity with probability $1 - \delta$:*

$$V^{\pi^*} - V^{\hat{\pi}^*} \le \underbrace{HR_{\max}C_T(\mathcal{F}_T, \pi^*)\sqrt{\frac{c_T}{N_o}\log\left(\frac{H}{\delta}\mathcal{N}_{\mathcal{F}_T}\left(\frac{1}{N_o}\right)\right)}}_{\text{transition term } \epsilon_T} + \underbrace{2\alpha\kappa C_R(\mathcal{F}_R, \pi^*)\sqrt{\frac{c_R}{N_p}\log\left(\frac{1}{\delta}\mathcal{N}_{\mathcal{F}_R}\left(\frac{1}{N_p}\right)\right)}}_{\text{reward term}},$$

*where $\alpha = 1$ for uniform sampling or $\alpha \le 1$ for uncertainty sampling, $C_R$ is a concentrability measure for the reward function, $\kappa = \sup_{r \in [-R_{\max}, R_{\max}]} \frac{1}{\sigma'(r)}$ measures the degree of non-linearity of the link function, and $c_T, c_R$ are universal constants.*

In the special case where both the transition and reward functions are learned on a fixed initial preference dataset (no preference elicitation; $|\mathcal{D}_{\text{offline}}| = 2N_p$), we recover Theorem 1 from Zhan et al. (2023a). Importantly, the coefficient $\alpha$ allows us to motivate the superior efficiency of uncertainty sampling over uniform sampling, observed empirically in Shin et al. (2022) and in our own experiments (Section 7). Uncertainty sampling learns accurate reward models with fewer preference queries when $\alpha < 1$, but can perform like uniform sampling in harder problems ($\alpha = 1$).

## 6 PREFERENCE ELICITATION FROM SIMULATED TRAJECTORIES

We now propose our alternative strategy for generating trajectories for offline preference elicitation: **Simulated Offline Preference-based Reward Learning (Sim-OPRL)**. This method simulates trajectories $(\tau_1, \tau_2)$ by leveraging the *learned environment model*. This overcomes a limitation of OPRL, which is only designed to reduce uncertainty about the reward functions in $\mathcal{R}$, by instead reducing uncertainty about which policies are plausibly optimal. Our approach is inspired by efficient online preference elicitation algorithms (Saha et al., 2023; Chen et al., 2022), which we modify for practical implementation. We account for the offline nature of our problem by avoiding regions that are out of the distribution of the data: the sampling strategy is *optimistic* with respect to uncertainty in rewards, but *pessimistic* with respect to uncertainty in transitions.

We summarize our approach to generating simulated trajectories for preference elicitation in Algorithm 2. First, we construct a set of candidate optimal policies $\Pi_{\text{offline}}$, containing policy $\pi^*_{\text{offline}}$ (optimal policy under the pessimistic model and the true reward function) with high probability – as demonstrated in Appendix A.5.2. Next, within this set of candidate policies, we identify the two most exploratory policies $\pi_1, \pi_2$, chosen to maximize preference uncertainty $u_{P_R}$. Finally, we roll out these policies within our learned transition model to generate a trajectory pair $(\tau_1, \tau_2)$ for preference feedback.

---

**Algorithm 2** Preference Elicitation through Simulated Trajectory Sampling.

---

**Input:** Pessimistic transition model $\hat{T}_{\text{inf}}$. Reward confidence set $\mathcal{R}$ and preference uncertainty function $u_{P_R}$.
**Output:** $(\tau_1, \tau_2)$
1: Estimate optimal offline policy set: $\Pi_{\text{offline}} = \{\pi \mid \exists \tilde{R} \in \mathcal{R} : \pi = \text{argmax}_{\pi \in \Pi} \mathbb{E}_{\tau \sim d^\pi_{\hat{T}_{\text{inf}}}}[\tilde{R}(\tau)]\}$
2: Identify exploratory policies: $\pi_1, \pi_2 = \text{argmax}_{\pi_1, \pi_2 \in \Pi_{\text{offline}}} \mathbb{E}_{\tau_1 \sim d^{\pi_1}_{\hat{T}_{\text{inf}}}, \tau_2 \sim d^{\pi_2}_{\hat{T}_{\text{inf}}}}[u_{P_R}(\tau_1, \tau_2)]$
3: Rollouts in model: $\tau_1 \sim d^{\pi_1}_{\hat{T}_{\text{inf}}}, \tau_2 \sim d^{\pi_2}_{\hat{T}_{\text{inf}}}$.

---

We first provide a theoretical analysis of the performance of Sim-OPRL, before proposing a practical implementation of our entire preference elicitation and policy optimization algorithm.

### 6.1 THEORETICAL GUARANTEES

We decompose suboptimality in a similar way to Section 5.1, but obtain a reward suboptimality term that depends on the learned dynamics model instead of the true one, and on $\pi^*_{\text{offline}}$ instead of $\pi^*$:

$$V^{\pi^*} - V^{\hat{\pi}^*} \le \underbrace{(V^{\pi^*}_{T,R} - V^{\pi^*}_{\hat{T}_{\text{inf}},R})}_{\text{transition term } \epsilon_T} + \underbrace{(V^{\pi^*_{\text{offline}}}_{\hat{T}_{\text{inf}},R} - V^{\pi^*_{\text{offline}}}_{\hat{T}_{\text{inf}},\hat{R}_{\text{inf}}})}_{\text{reward term}}. \tag{4}$$

Analysis of the suboptimality due to transition error is identical to above, but the reward term is thus significantly different. By design, our sampling strategy ensures good coverage of preferences over $\pi^*_{\text{offline}}$ within the learned environment model, which **eliminates the concentrability term for the reward** $C_R$. We refer the reader to Appendix A.5 for the proof of Theorem 6.1.

**Theorem 6.1.** *For any* $\delta \in (0,1]$, *let* $\beta_T = c_T^{\text{MLE}} \log(H\mathcal{N}_{\mathcal{F}_T}(1/N_o)/\delta)/N_o$ *and* $\beta_R = c_R^{\text{MLE}} \log(\mathcal{N}_{\mathcal{F}_R}(1/N_p)/\delta)/N_p$, *where* $N_o = H|\mathcal{D}_{\text{offline}}|$ *is the number of observed transitions in the observational dataset and* $c_T^{\text{MLE}}, c_R^{\text{MLE}}$ *are universal constants. The policy* $\hat{\pi}^*$ *estimated by Algorithm 1, with a preference sampling strategy based on rollouts in the learned transition model, achieves the following suboptimality with probability* $1 - \delta$:

$$V^{\pi^*} - V^{\hat{\pi}^*} \leq \underbrace{HR_{\max}C_T(\mathcal{F}_T, \pi^*)\sqrt{\frac{c_T}{N_o}\log\left(\frac{H}{\delta}\mathcal{N}_{\mathcal{F}_T}\left(\frac{1}{N_o}\right)\right)}}_{\text{transition term } \epsilon_T} + \underbrace{2\kappa\sqrt{\frac{c_R}{N_p}\log\left(\frac{1}{\delta}\mathcal{N}_{\mathcal{F}_R}\left(\frac{1}{N_p}\right)\right)}}_{\text{reward term}}.$$

*where* $\kappa = \sup_{r \in [-R_{\max}, R_{\max}]} \frac{1}{\sigma'(r)}$ *measures the degree of non-linearity of the link function, and* $c_T, c_R$ *are universal constants.*

## 6.2 DISCUSSION

Our theoretical results demonstrate that the learned policy can achieve performance comparable to the optimal policy, and thus satisfy our optimality criterion in Definition 3.1, provided it is covered by the offline data ($C_T(\mathcal{F}_T, \pi^*), C_R(\mathcal{F}_R, \pi^*) < \infty$). Following our analysis, a suboptimal dataset requires more preferences to achieve a certain policy performance, as the concentrability terms $C_T$ or $C_R$ are large. Empirical results in Section 7 confirm that sample efficiency is worse when the behavioral policy is more suboptimal.

**Offline Trajectories vs. Simulated Rollouts.** While both OPRL and Sim-OPRL depend on the offline dataset for estimating environment dynamics, they induce different suboptimality in modeling preference feedback. Simulated rollouts are designed to achieve good coverage of the optimal offline policy $\pi^*_{\text{offline}}$, which avoids wasting preference budget on trajectories with low rewards or high transition uncertainty. In contrast, as shown in Zhan et al. (2023a), due to the dependence of preferences on full trajectories, the reward concentrability term $C_R$ in Theorem 5.1 can be very large. While sampling from the offline buffer is not sensitive to the quality of the transition model, good coverage of the optimal policy is needed from both transition and preference data to achieve low suboptimality.

**Transition vs. Preference Model Quality.** Our theoretical analysis also suggests an interesting trade-off in the sample efficiency of our approach, depending on the accuracy of the transition model. The width of the confidence interval reduces as significance parameter $\delta$ or dataset size increase, or as function class complexity $\mathcal{N}_{\mathcal{F}_T}$ decreases. For a target suboptimality gap $\epsilon$, provided the optimal offline policy $\pi^*_{\text{offline}}$ has a gap $\epsilon_T < \epsilon$, then the number of preferences required is of the order of $\mathcal{O}(\log(1/\delta)/(\epsilon - \epsilon_T)^2)$. A more accurate transition model should therefore require fewer preference samples to achieve a given suboptimality, which we again confirm empirically.

## 6.3 PRACTICAL IMPLEMENTATION

We now complete the general algorithmic framework discussed above with a possible implementation strategy, allowing for empirical validation. In fact, with minor changes to the following framework, our paper also proposes a feasible implementation of related theoretical algorithms (Chen et al., 2022; Zhan et al., 2023a). We refer the reader to Appendix C for further detail.

**Model Learning and Policy Optimization.** Following prior work in offline reinforcement learning (Yu et al., 2020), we train ensembles of $N_T$ and $N_R$ neural network models for the transition and reward functions on different bootstraps of the data (Lakshminarayanan et al., 2017), denoted $\{\hat{T}_1, \dots \hat{T}_{N_T}\}$ and $\{\hat{R}_1, \dots \hat{R}_{N_R}\}$. We estimate MLE and uncertainty functions as follows:

$$\hat{T}(\cdot|s,a) = \frac{1}{N_T}\sum_{i=1}^{N_T}\hat{T}_i(\cdot|s,a); \quad u_T(s,a) = \max_{i,j\in[\![1,N_T]\!]}|\hat{T}_i(\cdot|s,a) - \hat{T}_j(\cdot|s,a)|_1 \cdot R_{\max}$$

$$\hat{R}(s,a) = \frac{1}{N_R}\sum_{i=1}^{N_T}\hat{R}_i(s,a); \quad u_R(s,a) = \max_{i,j\in[\![1,N_R]\!]}|\hat{R}_i(s,a) - \hat{R}_j(s,a)|_1$$

Each $\hat{R}_i$ in the ensemble has an associated preference function defined by the Bradley-Terry model, with $\sigma$ as the sigmoid function. We obtain preference uncertainty through variation over the ensemble as in Equation (2). Recall that transition and reward models are trained on $\mathcal{D}_{\text{offline}}$ and $\mathcal{D}_{\text{pref}}$ respectively; for computational efficiency, we sample preferences in batches of to reduce the number of reward model updates needed.

We approximate the pessimistic objective in Equation (3) by penalizing the reward function with the uncertainty, as in Lagrangian formulations of model-based offline RL (Yu et al., 2020; Rigter et al., 2022). We solve for the following objective with a traditional reinforcement learning algorithm:

$$\hat{\pi}^* = \text{argmax}_{\pi \in \Pi} \mathbb{E}_{(s,a) \sim d_{\hat{T}}^\pi} [\hat{R}(s,a) - \lambda_R u_R(s,a) - \lambda_T u_T(s,a)], \tag{5}$$

where hyperparameters $\lambda_T, \lambda_R$ control the degree of conservatism. Note that in our theoretical analysis, this was achieved through parameters $\beta_T, \beta_R$ which affect the width of the confidence intervals $u_R$ and $u_T$, but their exact value cannot be estimated. We show in Appendix A.2 that Equation (5) indeed lower bounds the true value function, under well-calibrated uncertainty estimates.

**Near-Optimal Policy Set and Exploratory Policies.** Sim-OPRL requires constructing $\Pi_{\text{offline}}$, a set of near-optimal policies within a pessimistic model of the environment. Following Lindner et al. (2021), we obtain a policy model for each element $\hat{R}_i$ of the reward ensemble. Policy models are optimized to maximize returns under the transition model $\hat{T}$ and the reward function $\hat{R}_i - \lambda_T u_T$, ensuring pessimism w.r.t transitions. Next, the most exploratory policies are identified by generating rollouts of each candidate policy within the learned model $\hat{T}$. The trajectories $(\tau_1, \tau_2)$ induced by different policies and maximizing the preference uncertainty function $u_{P_R}(\tau_1, \tau_2)$ are used for preference feedback. We refer the reader to Appendix C for further detail.

# 7 EXPERIMENTAL RESULTS

In this section, we demonstrate the effectiveness of our preference elicitation strategy, Sim-OPRL, across a range of offline reinforcement learning environments and datasets. We demonstrate its **superior performance over OPRL, as expected from our theoretical analysis**.

Since our closest related works do not propose any experimental validation (Chen et al., 2022; Zhan et al., 2023a), we propose a practical implementation of Preference-based Optimistic Planning (PbOP) in Appendix C; this elicitation method queries feedback over trajectory rollouts in the *true environment* (Chen et al., 2022). We also compare against OPRL (Shin et al., 2022) with uniform and uncertainty-sampling. Finally, we report the performance of $\pi_{\text{offline}}^*$ and $\pi^*$ as upper bounds for the performance of our algorithm: the former is trained in the learned transition model with access to the true reward, and the latter has full knowledge of both transition and reward function.

We compare different preference elicitation strategies on a range of environments detailed in Appendix D. Among others, we explore environments from the D4RL benchmark (Fu et al., 2020) identified as particularly challenging offline preference-based reinforcement learning tasks (Shin et al., 2022), as well as a medical simulation designed to model the evolution of patients with sepsis (Oberst and Sontag, 2019). As detailed in Appendix D, these environments consist of high-dimensional state spaces with continuous or discrete action spaces, follow complex transition dynamics, and have sparse or non-linear rewards and termination conditions. This makes them representative of the challenge of learning a reward function and learning offline in a real-world application. In particular, the sepsis simulation environment is commonly used in medically-motivated offline RL work (Tang and Wiens, 2021; Pace et al., 2023), and highlights another advantage of Sim-OPRL over OPRL: it does not require feedback on real trajectories from the observational dataset. In a sensitive setting such as healthcare where data access is carefully controlled, it may be attractive to query experts about *synthetic* trajectories rather than real samples.

**Performance against State-of-the-Art.** Performance and sample complexity results with different preference elicitation methods are given in Figure 1 and Table 2. Within the offline approaches, Sim-OPRL consistently achieves better environment returns than OPRL with much fewer preference queries. In line with our theoretical analysis, our empirical results therefore demonstrate that policy-based sampling in Sim-OPRL is more efficient than maximizing information gain on the reward function (uncertainty-based OPRL), which echoes similar conclusions reached in prior work on online preference elicitation (Lindner et al., 2021; Chen et al., 2022).

Table 2: **Sample complexity $N_p$ under different preference elicitation strategies**, to reach a suboptimality gap of $\epsilon = 20$ over normalized returns. Mean and 95% confidence interval over 6 experiments. The best-performing offline method is highlighted in bold, ✗ marks when the target suboptimality could not be achieved. Note that PbOP has an advantage by having access to direct interaction with the environment.

| Environment | OPRL Uniform | OPRL Uncertainty | **Sim-OPRL (Ours)** | PbOP (Online) |
|---|---|---|---|---|
| Star MDP | $32 \pm 4$ | $30 \pm 4$ | $\mathbf{4 \pm 2}$ | $4 \pm 2$ |
| Gridworld | $105 \pm 11$ | $66 \pm 7$ | $\mathbf{49 \pm 7}$ | $32 \pm 4$ |
| MiniGrid-FourRooms | $92 \pm 7$ | $53 \pm 5$ | $\mathbf{41 \pm 5}$ | $25 \pm 3$ |
| HalfCheetah-Random | $108 \pm 9$ | $71 \pm 8$ | $\mathbf{50 \pm 10}$ | $36 \pm 3$ |
| Sepsis Simulation | ✗ | $642 \pm 72$ | $\mathbf{225 \pm 46}$ | $75 \pm 11$ |

(a) StarMDP.  (b) Gridworld.  (c) MiniGrid-FourRooms.

(d) HalfCheetah-Random.  (e) Sepsis Simulation.

Figure 1: **Environment returns under different preference elicitation strategies.** Mean and 95% confidence interval over 6 experiments. Environment returns are normalized between 0 and 100. Only OPRL and Sim-OPRL are fully offline.

As an upper bound for the performance of our algorithm, we include baselines that have access to the environment: we report the performance of the optimal policy $\pi^*$, as well as that of an algorithm querying feedback over optimistic rollouts in the *real* environment (Chen et al., 2022, PbOP). In Figure 1, the PbOP method naturally reaches a superior policy with fewer samples as it allows environment interaction and can thus improve its estimate of the transition model in parallel to learning the preference function. As supported by our theoretical analysis, this result stresses the importance of having a high-quality transition model to make our method effective. We explore this in more detail in our following ablations.

**Algorithm Ablations.** We conduct ablations for our algorithm on a simple tabular MDP, with results in Figure 2. This example (transition and reward details deferred to Appendix D) illustrates the importance of pessimism with respect to the transition model. Even with access to true rewards, $\pi^*_{\text{offline}}$ is pessimistic to avoid the out-of-distribution state, as it is unclear how to reach it. Thus, in Figure 2, we see a drop in performance if pessimism is not applied to the output policy (purple lines). This confirms the theoretical insights from Zhu et al. (2023); Zhan et al. (2023a), who demonstrate the importance of pessimism in offline preference-based RL problems. Pessimism is also crucial in simulated rollouts, to

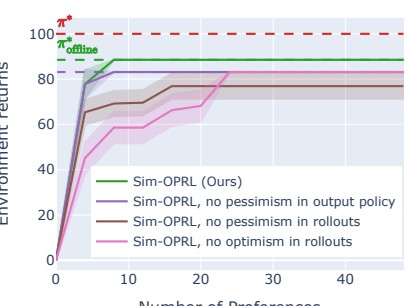

Figure 2: **Algorithm ablations** (StarMDP).

avoid wasting preference budget on regions of low confidence — as value estimates are inaccurate in any case. This is reflected in lower performance without pessimism w.r.t the transition model

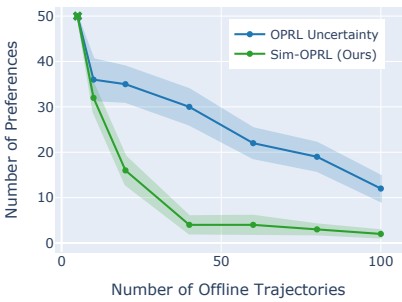

(a) As a function of offline dataset size.

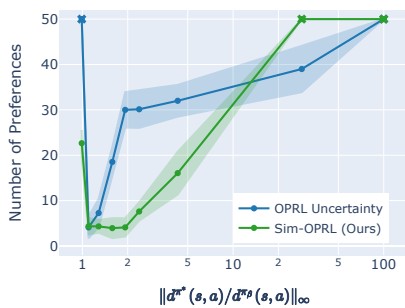

(b) As a function of dataset optimality.

Figure 3: **Preference sample complexity $N_p$ as function of the properties of the observational data**, to reach a suboptimality gap of $\epsilon = 20$ over normalized environment returns (Star MDP). Mean and 95% confidence intervals over 6 experiments. $\times$ marks when the target suboptimality could not be achieved.

in Figure 2 (brown line), and which **could be seen as the naive adaptation of online preference elicitation methods to our setting** (Chen et al., 2022; Lindner et al., 2022). We also note the importance of optimism with respect to the reward uncertainty, both in OPRL in Figure 1 and in our own model-based rollouts in Figure 2.

**Transition vs. Preference Model Quality.** Next, we empirically study the trade-off between transition and preference model performance in our problem setting. Still in the Star MDP, in the low-data regime, the error $\epsilon_T$ incurred in estimating the value function due to the misspecification of the transition model is large. As dictated by our theoretical analysis and as visualized in Figure 3a, this significantly increases the number of preference samples $N_p$ required to achieve good final performance. At the other end of the spectrum, if the offline dataset is large and allows modeling the transition model accurately, then $\epsilon_T$ is small and the number of preference samples $N_p$ needed shrinks. We observe a similar trend for both Sim-OPRL and our OPRL uncertainty-sampling baseline.

We also measure how the coverage of the optimality of the dataset affects performance in our setting. In Figure 3b, we vary the behavioral policy $\pi_\beta$ underlying the offline data, ranging from optimal (density ratio coefficient $= 1$) to highly suboptimal (large density ratio coefficient). The concentrability terms $C_T$ and $C_R$ are challenging to measure as they require considering entire function classes, but we report the accuracy of the maximum likelihood estimate for both models in Appendix E. We observe that preference elicitation methods perform best when the data is close to optimal (with the exception of a fully optimal, non-diverse dataset making reward learning from preferences challenging). More preference samples are required if the observational dataset has poor coverage of the optimal policy (large $C_T(\mathcal{F}_T, \pi^*)$), as the transition and reward models become less accurate for the trajectory distribution of interest. We also validate this conclusion on HalfCheetah datasets of varying optimality in Appendix E.

## 8   Conclusion

Our work shows the potential of integrating human feedback within the framework of offline RL. We address the challenges of preference elicitation in a fully offline setup by exploring two key methods: sampling from the offline dataset (Shin et al., 2022, OPRL) and generating model rollouts (Sim-OPRL). By employing a pessimistic approach to handle out-of-distribution data and an optimistic strategy to acquire informative preferences, Sim-OPRL balances the need for robustness and informativeness in learning an optimal policy.

We provide theoretical guarantees on the sample complexity of both approaches, demonstrating that performance depends on how well the offline data covers the optimal policy. Empirical evaluations on various environments confirm the practical effectiveness of our algorithm, as Sim-OPRL consistently outperforms OPRL baselines in all settings.

Overall, our approach not only advances the state-of-the-art in offline preference-based RL but also takes a significant step toward improving the practical utility of offline RL. This opens up new avenues for real-world applications of RL in healthcare, robotics, and manufacturing, where interaction with the environment is challenging but domain experts can be queried for feedback. Looking forward, a natural extension will be to explore alternative sources of information from experts, still without direct environment interaction.

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

# A    THEORETICAL DETAILS

This appendix provides proofs for the presented theorems and lemmas. In subsection A.1, we provide details on how we define the maximum likelihood estimators and confidence intervals of the preference and transition models. In subsection A.2 we provide the proof that our uncertainty-penalized objective in Equation (5) lower bounds the true value function and thus forms a valid pessimistic framework. In Appendix A.3, we show that the suboptimality of our offline preference elicitation framework is not lower-bounded by the performance of the optimal offline policy. In Appendix A.4, we provide our proof of theorem 5.1, analyzing the suboptimality of preferences sampled from an offline dataset. Finally, in Appendix A.5, we prove Theorem 6.1, which analyzes the suboptimality of preference sampling over simulated rollouts.

## A.1    MAXIMUM LIKELIHOOD AND CONFIDENCE INTERVALS

Let $\mathcal{F}_g$ denote a function class over $\mathcal{X} \to \Delta_{\mathcal{Y}}$, where $\mathcal{X}, \mathcal{Y}$ are measurable sets, and $g \in \mathcal{F}_g$ denotes a function to be estimated.

Let $\hat{g}$ denote the maximum likelihood estimator (MLE) of $g$ based on a dataset $\mathcal{D} = \{(x^n, y^n)\}_{n=1}^N$: $\hat{g} = \operatorname{argmax}_{\tilde{g} \in \mathcal{F}_g} \mathbb{E}_{(x,y) \sim \mathcal{D}} \log(\tilde{g}(y|x))$. We construct the confidence set around the MLE as follows:

$$\mathcal{G} = \{\tilde{g} \in \mathcal{F}_g \mid \mathbb{E}_{(x,y) \sim \mathcal{D}} \left[ \log \frac{\hat{g}(y|x)}{\tilde{g}(y|x)} \right] \leq \beta\}$$

**Lemma A.1** (MLE Guarantee, Lemma 1 in Zhan et al. (2023a)). *Let $\delta \in (0,1]$ and define the event $\mathcal{E}$ that $g \in \mathcal{G}$. If*

$$\beta = \frac{c_{MLE}}{N} \log \left( \frac{1}{\delta} \mathcal{N}_{\mathcal{F}_g} \left( \frac{1}{N} \right) \right),$$

*where $c_{MLE} > 0$ is a universal constant, then $P(\mathcal{E}) \geq 1 - \delta/2$.*

*Proof.* The proof follows that of Lemma 1 in Zhan et al. (2023a) and uses Cramér-Chernoff's method.

Let $\bar{\mathcal{B}}$ be a $1/N$-bracket of $\mathcal{F}_g$ with $|\bar{\mathcal{B}}|_1 = \mathcal{N}_{\mathcal{F}_g}(1/N)$. Denote the set of all right brackets in $\bar{\mathcal{B}}$ by $\tilde{\mathcal{B}} = \{b : \exists b' s.t. [b', b] \in \bar{\mathcal{B}}\}$. For $b \in \tilde{\mathcal{B}}$, we have:

$$\mathbb{E} \left[ \exp \left( \sum_{n=1}^N \log \frac{b(y^n|x^n)}{g(y^n|x^n)} \right) \right] = \prod_{n=1}^N \mathbb{E} \left[ \exp \left( \log \frac{b(y^n|x^n)}{g(y^n|x^n)} \right) \right]$$

$$= \prod_{n=1}^N \mathbb{E} \left[ \frac{b(y^n|x^n)}{g(y^n|x^n)} \right]$$

$$= \prod_{n=1}^N \mathbb{E} \left[ \sum_y b(y|x^n) \right]$$

$$\leq (1 + 1/N)^N \leq e.$$

as samples in $\mathcal{D}$ as i.i.d. We use the Tower property in the third step and the fact that $b$ is a $1/N$-bracket for $\mathcal{F}_g$ in the fourth: there exists $g' \in \mathcal{F}_g$ such that $\|g(\cdot|x) - b(\cdot|x)\|_1 \leq 1/N$ and thus $\|b(\cdot|x)\|_1 \leq 1 + 1/N$, for all $x \in \mathcal{X}$.

Then by Markov's inequality, for any $\delta \in (0,1]$, we have:

$$P \left( \sum_{n=1}^N \log \frac{b(y^n|x^n)}{g(y^n|x^n)} > \log(\delta) \right) \leq \mathbb{E} \left[ \exp \left( \sum_{n=1}^N \log \frac{b(y^n|x^n)}{g(y^n|x^n)} \right) \right] \cdot \exp(-\log(1/\delta))$$

$$\leq e\delta.$$

By union bound, we have for all $b \in \tilde{\mathcal{B}}$,

$$P \left( \sum_{n=1}^N \log \frac{b(y^n|x^n)}{g(y^n|x^n)} > c_{MLE} \log \left( \frac{1}{\delta} \mathcal{N}_{\mathcal{F}_g} \left( \frac{1}{N} \right) \right) \right) \leq \delta/2,$$

where $c_{MLE} > 0$ is a universal constant.

Finally, for all $\tilde{g} \in \mathcal{F}_g$, there exists $b \in \tilde{\mathcal{B}}$ such that $g(\cdot|x) \leq \tilde{g}(\cdot|x)$ for all $x \in \mathcal{X}$. As a result, for all $\tilde{g} \in \mathcal{F}_g$, we have:

$$
P\left(\sum_{n=1}^{N} \log \frac{\tilde{g}(y^n|x^n)}{g(y^n|x^n)} > c_{MLE} \log\left(\frac{1}{\delta}\mathcal{N}_{\mathcal{F}_g}\left(\frac{1}{N}\right)\right)\right) \leq \delta/2.
$$

$\square$

Under this event $\mathcal{E}$, we have $g \in \mathcal{G}$ with probability $1 - \delta/2$. A confidence interval constructed via loglikelihood also incurs a bound on the total variation (TV) distance between $g$ and $\tilde{g} \in \mathcal{G}$:

**Lemma A.2** (TV-distance to MLE). *Under the event $\mathcal{E}$, we have, with probability $1 - \delta$, for all $\tilde{g} \in \mathcal{G}$:*

$$
\mathbb{E}_{x \sim \mathcal{D}}\left[\|g(\cdot|x) - \tilde{g}(\cdot|x)\|_1^2\right] \leq \frac{c}{N}\log\left(\frac{1}{\delta}\mathcal{N}_{\mathcal{F}_g}\left(\frac{1}{N}\right)\right), \tag{6}
$$

*where $c > 0$ is a universal constant.*

*Proof.* The proof follows that of Liu et al. (2022a), Proposition 14. $\square$

This guarantees that the true function is within an interval around the MLE estimate with high probability.

We apply these lemmas to our MLE estimates of transition and reward functions in Algorithm 1 to obtain the following guarantees.

Let $\mathcal{E}_R$ denote the event $R \in \mathcal{R}$ and $\mathcal{E}_T$ denote the event $T \in \mathcal{T}$, $\mathcal{R}$ and $\mathcal{T}$ denote the respective confidence sets around the MLE. By Lemma A.1, events $\mathcal{E}_R$ and $\mathcal{E}_T$ have probability $1 - \delta/2$ if we choose $\beta_R = c_R^{\text{MLE}}\log(\mathcal{N}_{\mathcal{F}_R}(1/N_p)/\delta)/N_p$ and $\beta_T = c_T^{\text{MLE}}\log(H\mathcal{N}_{\mathcal{F}_T}(1/N_o)/\delta)/N_o$, where $c_R^{\text{MLE}}, c_T^{\text{MLE}}$ are universal constants.

## A.2 MODEL-BASED PESSIMISM AND UNCERTAINTY PENALTIES

**Lemma A.3** (Telescoping Lemma). *For any reward model $R \in \mathcal{F}_R$, and any two transition models $T, \hat{T} \in \mathcal{F}_T$:*

$$
V_{T,R}^{\pi} - V_{\hat{T},R}^{\pi} \leq \mathbb{E}_{\tau \sim d_{\hat{T}}^{\pi}}\left[\sum_{s_j, a_j \in \tau} \|T(\cdot|s_j, a_j) - \hat{T}(\cdot|s_j, a_j)\|_1\right] \cdot R_{max}
$$

*Proof.* The proof follows that of Lemma 4.1 in Yu et al. (2020) or Lemma 4 in Zhan et al. (2023a).

Let $W_j$ be the expected return under policy $\pi$, with transition model $\hat{T}$ for the first $j$ steps, then transition model $T$ for the rest of the episode. We have:

$$
V_{T,R}^{\pi} - V_{\hat{T},R}^{\pi} = \sum_{j=0}^{H-1} W_j - W_{j+1}.
$$

Now,

$$
W_j = R_j + \mathbb{E}_{s_j, a_j \sim \pi, \hat{T}}\left[\mathbb{E}_{s_{j+1} \sim T(\cdot|s_j, a_j)}[V_{T,R}^{\pi}(s_{j+1})]\right]
$$

$$
W_{j+1} = R_j + \mathbb{E}_{s_j, a_j \sim \pi, \hat{T}}\left[\mathbb{E}_{s_{j+1} \sim \hat{T}(s_j, a_j)}[V_{T,R}^{\pi}(s_{j+1})]\right]
$$

where $R_j$ is the expected return of the first $j$ steps taken in $\hat{T}$. Therefore,

$$
W_j - W_{j+1} = \mathbb{E}_{s_j, a_j \sim \pi, \hat{T}}\left[\mathbb{E}_{s_{j+1} \sim T(\cdot|s_j, a_j)}[V_{T,R}^{\pi}(s_{j+1})] - \mathbb{E}_{s_{j+1} \sim \hat{T}(s_j, a_j)}[V_{T,R}^{\pi}(s_{j+1})]\right]
$$

$$
\leq \mathbb{E}_{s_j, a_j \sim \pi, \hat{T}}\left[\|T(\cdot|s_j, a_j) - \hat{T}(\cdot|s_j, a_j)\|_1 \cdot R_{\max}\right]
$$

under the boundedness assumption for $R$. Finally, we have:

$$
\begin{aligned}
V_{T,R}^{\pi} - V_{\hat{T},R}^{\pi} &= \sum_{j=0}^{H-1} W_j - W_{j+1} \\
&= \sum_{j=0}^{H-1} \mathbb{E}_{s_j,a_j \sim \pi, \hat{T}} \left[ \mathbb{E}_{s_{j+1} \sim T(\cdot|s_j,a_j)}[V_{T,R}^{\pi}(s_{j+1})] - \mathbb{E}_{s_{j+1} \sim \hat{T}(s_j,a_j)}[V_{T,R}^{\pi}(s_{j+1})] \right] \\
&\leq \sum_{j=0}^{H-1} \mathbb{E}_{s_j,a_j \sim \pi, \hat{T}} \left[ \|T(\cdot|s_j,a_j) - \hat{T}(\cdot|s_j,a_j)\|_1 \cdot R_{\max} \right] \\
&= \mathbb{E}_{\tau \sim d_{\hat{T}}^{\pi}} \left[ \sum_{s_j,a_j \in \tau} \|T(\cdot|s_j,a_j) - \hat{T}(\cdot|s_j,a_j) \cdot R_{\max}\|_1 \right]
\end{aligned}
$$

$\square$

**Lemma A.4** (Pessimistic Transition Model). *Under event $\mathcal{E}_T$, for all $\pi \in \Pi, \tilde{R} \in \mathcal{F}_R$:*

$$
V_{\hat{T},\tilde{R}-u_T}^{\pi} \leq V_{T,\tilde{R}}^{\pi}.
$$

*Proof.*

$$
\begin{aligned}
V_{T,\tilde{R}}^{\pi} &= V_{\hat{T},\tilde{R}}^{\pi} - (V_{\hat{T},\tilde{R}}^{\pi} - V_{T,\tilde{R}}^{\pi}) \\
&\geq \mathbb{E}_{\tau \sim d_{\hat{T}}^{\pi}} \left[ \tilde{R}(\tau) \right] - \mathbb{E}_{\tau \sim d_{\hat{T}}^{\pi}}[u_T(\tau)] \\
&= \mathbb{E}_{\tau \sim d_{\hat{T}}^{\pi}} \left[ \tilde{R}(\tau) - u_T(\tau) \right]
\end{aligned}
$$

where we have used the telescoping lemma (Lemma A.3), and where $u_T(\tau) = \sum_{(s,a) \in \tau} u_T(s,a) \geq \sum_{(s,a) \in \tau} \|\hat{T}(\cdot|s,a) - T(\cdot|s,a)\|_1 \cdot R_{\max}$ under event $\mathcal{E}_T$. $\square$

**Lemma A.5** (Pessimistic Reward Model). *Under event $\mathcal{E}_R$, for all $\pi \in \Pi, \tilde{T} \in \mathcal{F}_T$:*

$$
V_{\tilde{T},\hat{R}-u_R}^{\pi} \leq V_{\tilde{T},R}^{\pi}.
$$

*Proof.*

$$
\begin{aligned}
V_{\tilde{T},R}^{\pi} &= V_{\tilde{T},\hat{R}}^{\pi} - (V_{\tilde{T},\hat{R}}^{\pi} - V_{\tilde{T},R}^{\pi}) \\
&= \mathbb{E}_{\tau \sim d_{\tilde{T}}^{\pi}} \left[ \hat{R}(\tau) \right] - \mathbb{E}_{\tau \sim d_{\tilde{T}}^{\pi}} \left[ \hat{R}(\tau) - R(\tau) \right] \\
&\geq \mathbb{E}_{\tau \sim d_{\tilde{T}}^{\pi}} \left[ \hat{R}(\tau) - u_R(\tau) \right]
\end{aligned}
$$

where we have used the fact that $|\hat{R}(\tau) - R(\tau)|_1 \leq \sum_{s,a \in \tau} |\hat{R}(s,a) - R(s,a)|_1 = \sum_{(s,a) \in \tau} u_R(s,a) = u_R(\tau)$ under event $\mathcal{E}_R$. $\square$

Combining the above two lemmas gives the following result:

**Corollary A.1.** *Under events $\mathcal{E}_T$ and $\mathcal{E}_R$, for all $\pi \in \Pi$:*

$$
V_{\hat{T},\hat{R}-u_T-u_R}^{\pi} \leq V_{T,R}^{\pi}.
$$

This justifies the overall objective considered in our pessimistic policy optimization procedure in Section 6.3.

### A.3 SUBOPTIMALITY LOWER BOUND: A COUNTEREXAMPLE

Let $\pi_{\text{offline}}^* = \operatorname{argmax}_{\pi \in \Pi} \min_{\tilde{T} \in \mathcal{T}} V_{\tilde{T},R}^{\pi}$ denote the optimal offline policy, which has access to the ground-truth reward function. In this section, we ask whether its suboptimality $\epsilon_T = V_{T,R}^{\pi^*} - V_{T,R}^{\pi_{\text{offline}}^*}$ is a lower bound for the suboptimality of our learned policy $\hat{\pi}^*$ after preference elicitation.

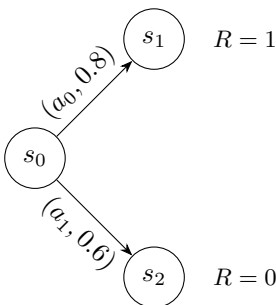

Figure 4: **Tabular MDP.** The environment starts in state $s_0$ and has horizon $H = 1$. Transition probabilities from state $s_0$ are given for the two binary actions $a_0, a_1$ (which send the agent to the other state with complementary probability).

**Counterexample.** Consider the MDP illustrated in Figure 4. The value function for the optimal policy $\pi^*$, which consists of taking action $a_0$, is: $V_{T,R}^{\pi^*} = 0.8 \cdot 1 + 0.2 \cdot 0 = 0.8$.

Now assume the following MLE estimate and uncertainty function for both the transition and reward models:

$$
\begin{aligned}
\hat{T}(s_1|s_0, a_0) = 0.5; && u_T(s_0, a_0) = 0.4 \\
\hat{T}(s_1|s_0, a_1) = 0.5; && u_T(s_0, a_1) = 0.1 \\
\hat{r}(s_1) = \hat{r}(s_2) = 0.5; && u_R(s_1) = u_R(s_2) = 0.5
\end{aligned}
$$

Assuming access to the learned transition model and the *true* reward function, we pessimistically estimate the value of both actions:

$$
\begin{aligned}
V_{\hat{T}_{\text{inf}}, R}^{a_0} &= 0.1 \cdot 1 + 0.9 \cdot 0 = 0.1 \\
V_{\hat{T}_{\text{inf}}, R}^{a_1} &= 0.6 \cdot 0 + 0.4 \cdot 1 = 0.4
\end{aligned}
$$

Thus, we have: $\Pi_{\text{offline}}^*(s_0) = \text{argmax}_a V_{\hat{T}_{\text{inf}}, R}^a = a_1$. The offline policy picks the suboptimal action since the worst-case returns of this action are lower than those estimated for $a_0$. Evaluating this policy in the real environment, we get $V_{T,R}^{\pi_{\text{offline}}^*} = 0.6 \cdot 0 + 0.4 \cdot 1 = 0.4$.

We now estimate the optimal policy in the learned transition and reward model. Applying pessimism with respect to both models, we get an equal estimated value of 0 for both actions $a_0$ and $a_1$. If policy optimization converges to $\hat{\pi}^* = a_0$, we achieve optimal performance with $V_{T,R}^{\hat{\pi}^*} = 0.8 > V_{T,R}^{\pi_{\text{offline}}^*}$.

This example demonstrates that $\epsilon_T = V_{T,R}^{\pi^*} - V_{T,R}^{\pi_{\text{offline}}^*} = 0.4$ is not a lower bound for the suboptimality of $\hat{\pi}^*$, as policy $\hat{\pi}^*$ can achieve better performance (even optimal performance in this case) if errors in transition and reward model estimation compensate each other.

## A.4 SUBOPTIMALITY OF OPRL: PROOF OF THEOREM 5.1

### A.4.1 SUBOPTIMALITY DECOMPOSITION

Recall that $\hat{T}_{\text{inf}}, \hat{R}_{\text{inf}} = \text{argmin}_{\tilde{T} \in \mathcal{T}, \tilde{R} \in \mathcal{R}} V_{\hat{T}, \hat{R}}^\pi$ denote the pessimistic transition and reward models, such that $\hat{\pi}^* = \text{argmax}_{\pi \in \Pi} V_{\hat{T}_{\text{inf}}, \hat{R}_{\text{inf}}}^\pi$. We have:

$$
\begin{aligned}
V^{\pi^*} - V^{\hat{\pi}^*} &= V_{T,R}^{\pi^*} - V_{T,R}^{\hat{\pi}^*} \\
&= (V_{T,R}^{\pi^*} - V_{\hat{T}_{\text{inf}}, \hat{R}_{\text{inf}}}^{\pi^*}) - (V_{T,R}^{\hat{\pi}^*} - V_{\hat{T}_{\text{inf}}, \hat{R}_{\text{inf}}}^{\pi^*}) \\
&\leq (V_{T,R}^{\pi^*} - V_{\hat{T}_{\text{inf}}, \hat{R}_{\text{inf}}}^{\pi^*}) - (V_{T,R}^{\hat{\pi}^*} - V_{\hat{T}_{\text{inf}}, \hat{R}_{\text{inf}}}^{\hat{\pi}^*}) \\
&\leq V_{T,R}^{\pi^*} - V_{\hat{T}_{\text{inf}}, \hat{R}_{\text{inf}}}^{\pi^*},
\end{aligned} \tag{7}
$$

where we have first used the optimality of $\hat{\pi}^*$ (stating that $V^{\hat{\pi}^*}_{\hat{T}_{\inf}, \hat{R}_{\inf}} \geq V^{\pi}_{\hat{T}_{\inf}, \hat{R}_{\inf}}$, for all $\pi$) and then the pessimism principle (stating that $V^{\hat{\pi}^*}_{\hat{T}_{\inf}, \hat{R}_{\inf}} \leq V^{\hat{\pi}^*}_{T, R}$).

Finally, we decompose the last term above as follows:

$$V^{\pi^*} - V^{\hat{\pi}^*} \leq \underbrace{(V^{\pi^*}_{T, \hat{R}_{\inf}} - V^{\pi^*}_{\hat{T}_{\inf}, \hat{R}_{\inf}})}_{\text{transition term}} + \underbrace{(V^{\pi^*}_{T, R} - V^{\pi^*}_{T, \hat{R}_{\inf}})}_{\text{reward term}} \tag{8}$$

We further analyze each term in the following sections.

### A.4.2 ANALYSIS OF THE TRANSITION TERM

In this section, we now upper bound the transition term defined in Equation (8).

**Lemma A.6** (Lemma 4, Zhan et al. (2023a)). *Under the event $\mathcal{E}_T$, with probability $1 - \delta$, we have for all $\tilde{T} \in \mathcal{T}$, for all $\tilde{R} \in \mathcal{G}_R$, for all $\pi$:*

$$\mathbb{E}_{d^{\pi}_T}[\tilde{R}(\tau)] - \mathbb{E}_{d^{\pi}_{\tilde{T}}}[\tilde{R}(\tau)] \leq H R_{max} C_T(\mathcal{F}_T, \pi) \sqrt{\frac{c_T}{N_o} \log\left(\frac{H}{\delta} \mathcal{N}_{\mathcal{F}_T}\left(\frac{1}{N_o}\right)\right)},$$

*where $c_T > 0$ is a constant.*

*Proof.* From the telescoping lemma (Lemma A.3), we have:

$$\begin{aligned}
V^{\pi}_{T, \tilde{R}} - V^{\pi}_{\tilde{T}, \tilde{R}} &\leq R_{\max} \mathbb{E}_{\tau \sim d^{\pi}_T}\left[\sum_{s_j, a_j \in \tau} \|T(\cdot|s_j, a_j) - \tilde{T}(\cdot|s_j, a_j)\|_1\right] \\
&\leq H R_{\max} \mathbb{E}_{(s,a) \sim d^{\pi}_T}\left[\|T(\cdot|s, a) - \tilde{T}(\cdot|s, a)\|_1\right] \\
&\leq H R_{\max} C_T(\mathcal{F}_T, \pi) \sqrt{\mathbb{E}_{(s,a) \sim D_{\text{offline}}}[\|T(\cdot|s, a) - \tilde{T}(\cdot|s, a)\|_1^2]}
\end{aligned}$$

Under event $\mathcal{E}_T$, by Lemma A.2, we have, with probability $1 - \delta$, for all $\tilde{T} \in \mathcal{T}$:

$$\mathbb{E}_{(s,a) \sim D_{\text{offline}}}[\|T(\cdot|s, a) - \tilde{T}(\cdot|s, a)\|_1^2] \leq \frac{1}{N_o} c_T \log\left(\frac{H}{\delta} \mathcal{N}_{\mathcal{F}_T}\left(\frac{1}{N_o}\right)\right)$$

This concludes our proof.

$\square$

### A.4.3 ANALYSIS OF THE REWARD TERM

Next, we upper bound the reward term defined in Equation (8).

As in Zhan et al. (2023a), we consider the following value function: $V^{\pi}_{T, R} = \mathbb{E}_{\tau \sim d^{\pi}_T}[R(\tau)] - \mathbb{E}_{\tau \sim d_{\text{pref}}}[R(\tau)]$, where $d_{\text{pref}}$ is a fixed reference trajectory distribution. This baseline subtraction, which doesn't affect either the optimal policy or the analysis of the transition term, is needed as the approximated confidence set is based on the uncertainty in *preference* between two trajectories, not in the reward of a single one.

**Definition A.1** (Preference concentrability coefficient). *The concentrability coefficient w.r.t. reward classes $\mathcal{F}_R$, a target policy $\pi^*$ and a reference trajectory distribution $d_{pref}$ is defined as:*

$$C_R(\mathcal{F}_R, \pi^*) = \frac{\mathbb{E}_{\tau_1 \sim d^{\pi^*}_T, \tau_2 \sim d_{\text{pref}}}[u_{P_R}(\tau_1, \tau_2)]}{\mathbb{E}_{\tau_1, \tau_2 \sim \mathcal{D}_{\text{offline}}}[u_{P_R}(\tau_1, \tau_2)]}$$

Note that, for the purpose of our analysis, our definition differs from that of Zhan et al. (2023a) who instead consider the max ratio of difference in rewards term: $|R(\tau_1) - R(\tau_2) - \tilde{R}(\tau_1) + \tilde{R}(\tau_2)|_1$ over the entire function class $\mathcal{F}_{\mathcal{R}}$.

**Lemma A.7.** *Let trajectories for preference elicitation be sampled uniformly from the offline dataset. Under the event $\mathcal{E}_R$, with probability $1 - \delta$, we have for all $\tilde{T} \in \mathcal{G}_T$, for all $\tilde{R} \in \mathcal{R}$, for all $\pi$:*

$$V_{T,R}^{\pi^*} - V_{T,\hat{R}_{inf}}^{\pi^*} \leq 2\alpha\kappa C_R(\mathcal{F}_R, \pi)\sqrt{\frac{c_R}{N_p}\log\left(\frac{1}{\delta}\mathcal{N}_{\mathcal{F}_R}\left(\frac{1}{N_p}\right)\right)},$$

*where $c_R > 0$ is a constant and $\kappa = \sup_{r\in[-R_{max},R_{max}]} \frac{1}{\sigma'(r)}$ measures the degree of non-linearity of the sigmoid function.*

*Proof.*

$$
\begin{aligned}
V_{T,R}^{\pi^*} - V_{T,\hat{R}_{inf}}^{\pi^*} &= \mathbb{E}_{\tau\sim d_T^{\pi^*}}[R(\tau)] - \mathbb{E}_{\tau\sim d_{\text{pref}}}[R(\tau)] - \mathbb{E}_{\tau\sim d_T^{\pi^*}}[\hat{R}_{\text{inf}}(\tau)] + \mathbb{E}_{\tau\sim d_{\text{pref}}}[\hat{R}_{\text{inf}}(\tau)] \\
&= \mathbb{E}_{\tau_1\sim d_T^{\pi^*},\tau_2\sim d_{\text{pref}}}[R(\tau_1) - R(\tau_2)] - (\hat{R}_{\text{inf}}(\tau_1) - \hat{R}_{\text{inf}}(\tau_2))] \\
&\leq \kappa\mathbb{E}_{\tau_1\sim d_T^{\pi^*},\tau_2\sim d_{\text{pref}}}[|P_R(\tau_1 \succ \tau_2) - P_{\hat{R}_{\text{inf}}}(\tau_1 \succ \tau_2)|_1] \\
&\leq \kappa\mathbb{E}_{\tau_1\sim d_T^{\pi^*},\tau_2\sim d_{\text{pref}}}[u_{P_R}(\tau_1,\tau_2)] \\
&= \kappa C_R(\mathcal{F}_R,\pi^*)\mathbb{E}_{\tau_1,\tau_2\sim\mathcal{D}_{\text{offline}}}[u_{P_R}(\tau_1,\tau_2)]
\end{aligned}
\tag{9}
$$

where $\kappa = \sup_{r\in[-R_{\max},R_{\max}]} \frac{1}{\sigma'(r)}$ measures the degree of non-linearity of the sigmoid function. In the first inequality, we have applied the mean value theorem, under Assumption 3.2. In the second inequality, we have used the definition of uncertainty function $u_{P_R}$ as we know $\hat{R}_{\text{inf}} \in \mathcal{R}$.

Now, under event $\mathcal{E}_R$, by Lemma A.2, we have, with probability $1 - \delta$ for all $\tilde{R} \in \mathcal{R}$:

$$\mathbb{E}_{(\tau_1,\tau_2)\sim\mathcal{D}_{\text{pref}}}[\|P_R(\tau_1 \succ \tau_2) - P_{\tilde{R}}(\tau_1 \succ \tau_2)\|_1^2] \leq \sqrt{\frac{c_R}{N_p}\log\left(\frac{1}{\delta}\mathcal{N}_{\mathcal{F}_R}\left(\frac{1}{N_p}\right)\right)}, \tag{10}$$

where $c_R > 0$ is a constant. This implies the following upper bound for the preference uncertainty function:

$$\mathbb{E}_{(\tau_1,\tau_2)\sim\mathcal{D}_{\text{pref}}}[u_{P_R}(\tau_1,\tau_2)] \leq 2\sqrt{\frac{c_R}{N_p}\log\left(\frac{1}{\delta}\mathcal{N}_{\mathcal{F}_R}\left(\frac{1}{N_p}\right)\right)} \tag{11}$$

Under uniform sampling, the distribution of preferences in $\mathcal{D}_{\text{pref}}$ is that of the offline dataset:

$$\mathbb{E}_{(\tau_1,\tau_2)\sim\mathcal{D}_{\text{offline}}}[u_{P_R}(\tau_1,\tau_2)] = \mathbb{E}_{(\tau_1,\tau_2)\sim\mathcal{D}_{\text{pref}}}[u_{P_R}(\tau_1,\tau_2)]$$

Thus,

$$V_{T,R}^{\pi^*} - V_{T,\hat{R}_{\text{inf}}}^{\pi^*} \leq 2\kappa C_R(\mathcal{F}_T,\pi)\sqrt{\frac{c_R}{N_p}\log\left(\frac{1}{\delta}\mathcal{N}_{\mathcal{F}_R}\left(\frac{1}{N_p}\right)\right)}.$$

$\square$

**Lemma A.8.** *Let trajectories for preference elicitation be sampled through uncertainty sampling from the offline dataset. Under the event $\mathcal{E}_R$, with probability $1 - \delta$, we have for all $\tilde{T} \in \mathcal{G}_T$, for all $\tilde{R} \in \mathcal{R}$, for all $\pi$:*

$$V_{T,R}^{\pi^*} - V_{T,\hat{R}_{inf}}^{\pi^*} \leq 2\alpha\kappa C_R(\mathcal{F}_T, \pi)\sqrt{\frac{c_R}{N_p}\log\left(\frac{1}{\delta}\mathcal{N}_{\mathcal{F}_R}\left(\frac{1}{N_p}\right)\right)},$$

*where $c_R > 0$ is a constant and $\alpha \leq 1$.*

*Proof.* The proof follows closely that of Lemma A.7. We introduce the preference concentrability coefficient defined for a general preference dataset:

$$C_R'(\mathcal{F}_T,\pi^*) = \frac{\mathbb{E}_{\tau_1\sim d_T^{\pi^*},\tau_2\sim d_{\text{pref}}}[u_{P_R}(\tau_1,\tau_2)]}{\mathbb{E}_{\tau_1,\tau_2\sim\mathcal{D}_{\text{pref}}}[u_{P_R}(\tau_1,\tau_2)]}$$

We start from Equation (9):

$$V_{T,R}^{\pi^*} - V_{T,\hat{R}_{\inf}}^{\pi^*} \le \kappa \mathbb{E}_{\tau_1 \sim d_T^{\pi^*}, \tau_2 \sim d_{\text{pref}}}[u_{P_R}(\tau_1, \tau_2)]$$
$$= \kappa C_R'(\mathcal{F}_R, \pi^*)\mathbb{E}_{\tau_1, \tau_2 \sim \mathcal{D}_{\text{pref}}}[u_{P_R}(\tau_1, \tau_2)]$$
$$\le 2\kappa C_R'(\mathcal{F}_R, \pi^*)\sqrt{\frac{c_R}{N_p}\log\left(\frac{1}{\delta}\mathcal{N}_{\mathcal{F}_R}\left(\frac{1}{N_p}\right)\right)}$$

where we have used Equation (11).

Now consider the dataset of uncertainty-sampled preferences $\mathcal{D}_{\text{pref}}$. By definition, we have:

$$\mathbb{E}_{\tau_1, \tau_2 \sim \mathcal{D}_{\text{pref}}}[u_{P_R}(\tau_1, \tau_2)] \ge \mathbb{E}_{\tau_1, \tau_2 \sim \mathcal{D}_{\text{offline}}}[u_{P_R}(\tau_1, \tau_2)]$$

Thus, we have: $C_R'(\mathcal{F}_R, \pi^*) \le C_R(\mathcal{F}_R, \pi^*)$. In other words, we can write: $C_R'(\mathcal{F}_R, \pi^*) = \alpha C_R(\mathcal{F}_R, \pi^*)$, where $\alpha \le 1$. This concludes our proof. $\square$

We now conclude the proof of Theorem 5.1 under events $\mathcal{E}_R$ and $\mathcal{E}_T$.

From Lemma A.6, we upper bound the transition term:

$$V_{T,\hat{R}_{\inf}}^{\pi^*} - V_{\hat{T}_{\inf}, \hat{R}_{\inf}}^{\pi^*} \le HR_{\max}C_T(\mathcal{F}_T, \pi^*)\sqrt{\frac{c_T}{N_o}\log\left(\frac{H}{\delta}\mathcal{N}_{\mathcal{F}_T}\left(\frac{1}{N_o}\right)\right)}$$

From Lemmas A.7 and A.8, we upper bound the reward term:

$$V_{T,R}^{\pi^*} - V_{T,\hat{R}_{\inf}}^{\pi^*} \le 2\alpha\kappa C_R(\mathcal{F}_R, \pi^*)\sqrt{\frac{c_R}{N_p}\log\left(\frac{1}{\delta}\mathcal{N}_{\mathcal{F}_R}\left(\frac{1}{N_p}\right)\right)},$$

where $\alpha = 1$ for uniform sampling or $\alpha \le 1$ for uncertainty sampling.

Combining with Equation (8), we obtain Theorem 5.1.

## A.5 SUBOPTIMALITY OF SIM-OPRL: PROOF OF THEOREM 6.1

### A.5.1 SUBOPTIMALITY DECOMPOSITION

We decompose the suboptimality slightly differently to Equation (7), introducing the optimal offline policy (optimal in the pessimistic model under the *true* reward function): $\pi_{\text{offline}}^* = \arg\max_{\pi \in \Pi} V_{\hat{T}_{\inf}, R}^{\pi}$.

$$V^{\pi^*} - V^{\hat{\pi}^*} = V_{T,R}^{\pi^*} - V_{T,R}^{\hat{\pi}^*}$$
$$= (V_{T,R}^{\pi^*} - V_{\hat{T}_{\inf}, \hat{R}_{\inf}}^{\pi_{\text{offline}}^*}) - (V_{T,R}^{\hat{\pi}^*} - V_{\hat{T}_{\inf}, \hat{R}_{\inf}}^{\pi_{\text{offline}}^*})$$
$$\le (V_{T,R}^{\pi^*} - V_{\hat{T}_{\inf}, \hat{R}_{\inf}}^{\pi_{\text{offline}}^*}) - (V_{T,R}^{\hat{\pi}^*} - V_{\hat{T}_{\inf}, \hat{R}_{\inf}}^{\hat{\pi}^*})$$
$$\le V_{T,R}^{\pi^*} - V_{\hat{T}_{\inf}, \hat{R}_{\inf}}^{\pi_{\text{offline}}^*}$$
$$= (V_{T,R}^{\pi^*} - V_{\hat{T}_{\inf}, R}^{\pi^*}) + (V_{\hat{T}_{\inf}, R}^{\pi^*} - V_{\hat{T}_{\inf}, \hat{R}_{\inf}}^{\pi_{\text{offline}}^*})$$
$$\le \underbrace{(V_{T,R}^{\pi^*} - V_{\hat{T}_{\inf}, R}^{\pi^*})}_{\text{transition term}} + \underbrace{(V_{\hat{T}_{\inf}, R}^{\pi_{\text{offline}}^*} - V_{\hat{T}_{\inf}, \hat{R}_{\inf}}^{\pi_{\text{offline}}^*})}_{\text{reward term}} \qquad (12)$$

where we have followed the same analysis as in Appendix A.4.1 and used the optimality of $\pi_{\text{offline}}^*$ in the last inequality.

The analysis of the transition term is identical to the above (Appendix A.4.2). We analyze the reward term next.

A.5.2 ANALYSIS OF THE REWARD TERM

**Lemma A.9** (Optimal Offline Policy In Set). *Let $\Pi_{offline}$ denote the following set of near-optimal pessimistic policies, under the pessimitic transition model $\hat{T}_{inf}$ and the reward confidence set $\mathcal{R}$:*

$$\Pi_{offline} = \{\pi \mid \pi = argmax_{\pi \in \Pi} \mathbb{E}_{\tau \sim d^\pi_{\hat{T}_{inf}}}[\tilde{R}(\tau)] \ \forall \tilde{R} \in \mathcal{R}\}$$

*Under event $\mathcal{E}_R$, we have $\pi^*_{offline} \in \Pi_{offline}$.*

*Proof.* Recall the definition of $\pi^*_{\text{offline}}$: $\pi^*_{\text{offline}} = \text{argmax}_{\pi \in \Pi} V^\pi_{\hat{T}_{\text{inf}}, R}$. Note that there is no need to consider the preference baseline term in $V^\pi$ when building $\Pi_{\text{offline}}$ since it is independent of the policy. Under event $\mathcal{E}_R$, we have $R \in \mathcal{R}$. Thus, $\pi^*_{\text{offline}} \in \Pi_{\text{offline}}$. $\qquad\square$

**Lemma A.10.** *Under event $\mathcal{E}_R$, we have, with probability $1 - \delta$:*

$$V^{\pi^*_{offline}}_{\hat{T}_{inf}, R} - V^{\pi^*_{offline}}_{\hat{T}_{inf}, \hat{R}_{inf}} \leq 2\kappa \sqrt{\frac{c_R}{N_p} \log\left(\frac{1}{\delta}\mathcal{N}_{\mathcal{F}_R}\left(\frac{1}{N_p}\right)\right)}$$

*Proof.*

$$
\begin{aligned}
V^{\pi^*_{\text{offline}}}_{\hat{T}_{\text{inf}}, R} &- V^{\pi^*_{\text{offline}}}_{\hat{T}_{\text{inf}}, \hat{R}_{\text{inf}}} \\
&= (V^{\pi^*}_{\hat{T}_{\text{inf}}, R} - V^{\pi^*}_{\hat{T}_{\text{inf}}, \hat{R}}) + (V^{\pi^*}_{\hat{T}_{\text{inf}}, \hat{R}} - V^{\pi^*}_{\hat{T}_{\text{inf}}, \hat{R}_{\text{inf}}}) \\
&= \mathbb{E}_{\tau \sim d^{\pi^*_{\text{offline}}}_{\hat{T}_{\text{inf}}}}[R(\tau)] - \mathbb{E}_{\tau \sim d_{\text{pref}}}[R(\tau)] - \mathbb{E}_{\tau \sim d^{\pi^*_{\text{offline}}}_{\hat{T}_{\text{inf}}}}[\hat{R}_{\text{inf}}(\tau)] + \mathbb{E}_{\tau \sim d_{\text{pref}}}[\hat{R}_{\text{inf}}(\tau)] \\
&= \mathbb{E}_{\tau_1 \sim d^{\pi^*_{\text{offline}}}_{\hat{T}_{\text{inf}}}, \tau_2 \sim d_{\text{pref}}}[R(\tau_1) - R(\tau_2)] - \mathbb{E}_{\tau_1 \sim d^{\pi^*_{\text{offline}}}_{\hat{T}_{\text{inf}}}, \tau_2 \sim d_{\text{pref}}}[\hat{R}_{\text{inf}}(\tau_1) - \hat{R}_{\text{inf}}(\tau_2)] \\
&\leq \kappa \mathbb{E}_{\tau_1 \sim d^{\pi^*_{\text{offline}}}_{\hat{T}_{\text{inf}}}, \tau_2 \sim d_{\text{pref}}}[P_R(\tau_1 \succ \tau_2) - P_{\hat{R}_{\text{inf}}}(\tau_1 \succ \tau_2)],
\end{aligned}
$$

where $\kappa = \sup_{r \in [-R_{\max}, R_{\max}]} \frac{1}{\sigma'(r)}$ measures the degree of non-linearity of the sigmoid function. We have applied the mean value theorem, under Assumption 3.2.

As $R_{\text{inf}} \in \mathcal{R}$, we have: $P_R(\tau_1 \succ \tau_2) - P_{\hat{R}_{\text{inf}}}(\tau_1 \succ \tau_2) \leq u_{P_R}(\tau_1, \tau_2)$.

Let $d_{\text{pref}}$ correspond to the distribution of the preference data, which consists of rollouts from exploratory policies within the learned environment model: $d_{\text{pref}} = d^{\pi_1}_{\hat{T}_{\text{inf}}}/2 + d^{\pi_2}_{\hat{T}_{\text{inf}}}/2$. Recall that the near-optimal policy set $\Pi_{\text{offline}}$ includes policy $\pi^*_{\text{offline}}$ (Lemma A.9) and that $\pi_1, \pi_2$ are the two more exploratory policies within this set:

$$\mathbb{E}_{\tau_1 \sim d^{\pi^*_{\text{offline}}}_{\hat{T}}, \tau_2 \sim d_{\text{pref}}}[u_{P_R}(\tau_1, \tau_2)] \leq \max_{\pi_1, \pi_2 \in \Pi_{\text{offline}}} \mathbb{E}_{\tau_1 \sim d^{\pi_1}_{\hat{T}}, \tau_2 \sim d^{\pi_2}_{\hat{T}}}[u_{P_R}(\tau_1, \tau_2)].$$

Now, under event $\mathcal{E}_R$, by Lemma A.2, we have, with probability $1 - \delta$ for all $\tilde{R} \in \mathcal{R}$:

$$\mathbb{E}_{(\tau_1, \tau_2) \sim \mathcal{D}_{\text{pref}}}[\|P_R(\tau_1 \succ \tau_2) - P_{\tilde{R}}(\tau_1 \succ \tau_2)\|_1^2] \leq \frac{c_R}{N_p} \log\left(\frac{1}{\delta}\mathcal{N}_{\mathcal{F}_R}\left(\frac{1}{N_p}\right)\right),$$

where $c_R > 0$ is a constant. This implies the following upper bound for the preference uncertainty function:

$$\mathbb{E}_{(\tau_1, \tau_2) \sim \mathcal{D}_{\text{pref}}}[u_{P_R}(\tau_1, \tau_2)] \leq 2\sqrt{\frac{c_R}{N_p} \log\left(\frac{1}{\delta}\mathcal{N}_{\mathcal{F}_R}\left(\frac{1}{N_p}\right)\right)}.$$

Thus, we obtain:

$$V^{\pi^*_{\text{offline}}}_{\hat{T}_{\text{inf}}, R} - V^{\pi^*_{\text{offline}}}_{\hat{T}_{\text{inf}}, \hat{R}_{\text{inf}}} \leq 2\kappa \sqrt{\frac{c_R}{N_p} \log\left(\frac{1}{\delta}\mathcal{N}_{\mathcal{F}_R}\left(\frac{1}{N_p}\right)\right)}.$$

The resulting sample complexity of $\mathcal{O}(\frac{\kappa^2 d}{\epsilon^2})$ matches that of active preference learning within a known environment (Saha et al., 2023; Chen et al., 2022).

$\qquad\square$

We now conclude the proof of Theorem 6.1 under events $\mathcal{E}_R$ and $\mathcal{E}_T$.

From Lemma A.6, we upper bound the transition term:

$$V_{T,R}^{\pi^*} - V_{\hat{T}_{\inf},R}^{\pi^*} \leq HR_{\max}C_T(\mathcal{F}_T, \pi^*)\sqrt{\frac{c_T}{N_o}\log\left(\frac{H}{\delta}\mathcal{N}_{\mathcal{F}_T}\left(\frac{1}{N_o}\right)\right)}.$$

From Lemma A.10, we upper bound the reward term:

$$V_{\hat{T}_{\inf},R}^{\pi^*_{\text{offline}}} - V_{\hat{T}_{\inf},\hat{R}_{\inf}}^{\pi^*_{\text{offline}}} \leq 2\kappa\sqrt{\frac{c_R}{N_p}\log\left(\frac{1}{\delta}\mathcal{N}_{\mathcal{F}_R}\left(\frac{1}{N_p}\right)\right)}.$$

Combining with Equation (12), we obtain Theorem 6.1.

## B  ADDITIONAL RELATED WORK

Table 3: **Comparison of broader related work on preference elicitation and preference-based RL.**

| Framework | Offline | Policy-based Sampling | Robustness Guarantees | Practical Implementation |
|---|---|---|---|---|
| PREFERENCE-BASED DEEP RL | | | | |
| D-REX (Brown et al., 2020) | ✓ | ✗ | ✗ | ✓ |
| PEBBLE (Lee et al., 2021) | ✗ | ✗ | ✗ | ✓ |
| MRN (Liu et al., 2022b) | ✗ | ✗ | ✗ | ✓ |
| SURF (Park et al., 2022) | ✗ | ✗ | ✗ | ✓ |
| PT (Kim et al., 2023) | ✗ | ✗ | ✗ | ✓ |
| IPL (Hejna and Sadigh, 2024) | ✓ | ✗ | ✗ | ✓ |
| PREFERENCE ELICITATION | | | | |
| PbOP (Chen et al., 2022) | ✗ | ✓ | ✓ | ✗ |
| REGIME (Zhan et al., 2023b) | ✗ | ✓ | ✓ | ✗ |
| FREEHAND (Zhan et al., 2023a) | ✓ | ✗ | ✓ | ✗ |
| OPRL (Shin et al., 2022) | ✓ | ✗ | ✗ | ✓ |
| MoP-RL (Liu et al., 2023) | ✗ | ✗ | ✗ | ✓ |
| Max. Regret (Wilde et al., 2020) | ✗ | ✓ | ✗ | ✓ |
| Biyik and Sadigh (2018) | ✗ | ✗ | ✓ | ✓ |
| Bıyık et al. (2019) | ✗ | ✗ | ✓ | ✓ |
| **Sim-OPRL (Ours)** | ✓ | ✓ | ✓ | ✓ |

In this section, we position this work against additional related literature. We summarize our analysis in Table 3.

First, we note that significant prior work on preference-based reinforcement learning is centered on improving preference modeling or policy optimization for a fixed (uncertainty-based or uniform) preference elicitation strategy (all methods in the top half of Table 3). This focus is orthogonal to our work, and could certainly be combined with our preference elicitation method for optimal performance.

Next, comparing with other preference elicitation methods in Table 3 (bottom half), we find that almost all works consider a **setting with online environment interaction**, which makes them incompatible with our problem setting. Online environment exploration eliminates the need for pessimism against transition dynamics (Levine et al., 2020), whereas we show that it is essential for both theoretical and empirical performance in the offline setting.

The uncertainty-based preference elicitation strategy adopted in most practical PbRL algorithms is to **minimize uncertainty about preferences**, or to maximize information gain about the reward function. From prior theoretical and empirical work (Chen et al., 2022; Lindner et al., 2021), we know that this strategy is often suboptimal compared to actively reducing the set of candidate optimal policies, and both our theoretical and experimental results confirm this.

Finally, note that ours is the first work to provide both a **theoretical and empirical analysis** of the sample complexity of preference elicitation in the fully offline setting.

**Beyond Offline Reinforcement Learning.**   With our community's recent surge in interest in preference-based reinforcement learning for its application to language model alignment (Stiennon et al., 2020; Ouyang et al., 2022), we foresee many possible applications and extensions of our work beyond offline RL. Sample efficiency in preference collection is of paramount importance in Reinforcement Learning from Human Feedback (RLHF): recent works consider active learning pipelines based on uncertainty sampling to improve efficiency (Chen et al., 2024; Melo et al., 2024), similar to OPRL. The insights from our work suggest that a model-based elicitation strategy like Sim-OPRL, designed to simulate responses from near-optimal policies, might perform even better. We believe this line of future work could prove critical to the language model alignment community.

## C   IMPLEMENTATION DETAILS

We trained all models on two 64-core AMD processors or a single NVIDIA RTX2080Ti GPU. The total wall-clock time for running all experiments presented in this paper amounted to less than 72 hours.

**Transition and Reward Function Training.**   For all baselines, transition and reward models were implemented as linear classifiers (for the Star MDP), as two-layer perceptions with ReLU activation and hidden layer dimension 32 (Gridworld, Sepsis, MiniGrid environments), or as 5-layer MLPs with ReLU activations and hidden sizes $[512, 256, 128, 64, 32]$ for the HalfCheetah environments. Training was carried out for two or one epochs for the transition and reward models respectively, with the Adam optimizer (Kingma and Ba, 2014) and a learning rate of $10^{-3}$.

We provide a more detailed practical algorithm for Sim-OPRL in Algorithm 3. For both our method and baselines relying on uncertainty sets (OPRL and PbOP), we estimated uncertainty sets by training models initialized with different random seeds on different bootstraps of the data (sampling 90% of the data with replacement). We consider ensembles of size $|\mathcal{T}| = |\mathcal{R}| = 5$ for both transition and reward models. Hyperparameters $\lambda_T, \lambda_R$ control the degree of pessimism in practice and could be considered equivalent to adjusting margin parameters $\beta_T, \beta_R$ in our conceptual algorithm proposed in Section 4. Since the exact values prescribed by our theoretical analysis cannot be estimated, the user must set these parameters themselves. Hyperparameter optimization in offline RL is a challenging problem (Levine et al., 2020); for our experiments, we simply set $\lambda_T = 0.5$, $\lambda_R = 0.1$ (StarMDP, Gridworld) and $\lambda_T = \lambda_R = 1$ for the Sepsis environment.

**Near-Optimal Policy Set and Exploratory Policies.**   Both Sim-OPRL and PbOP require constructing a set of near-optimal policies within a learned model of the environment. Note that the PbOP algorithm in Chen et al. (2022) proposes to construct the near-optimal policy set by considering all policies that have a preference greater than $1/2$ over *all other policies in* $\Pi$, under a transition and preference uncertainty bonus. This is infeasible to estimate in practice; we modified the algorithm to allow for practical implementation. The motivation in building the set of plausibly optimal policies remains the same, but the theoretical guarantees may not hold.

We build $\Pi_{\text{offline}}$ by maintaining a policy model for all $\tilde{R} \in \mathcal{R}$, i.e., each element of the reward ensemble. Policy models are optimized to maximize returns under the transition model $\hat{T}$ and the reward function $\tilde{R} - \lambda_T u_T$ (Sim-OPRL) or $\tilde{R} + \lambda_T u_T$ (PbOP). Next, the most exploratory policies are identified by generating 10 rollouts of each of the candidate policies within the learned (SimOPRL) or true (PbOP) model. The trajectories $(\tau_1, \tau_2)$ maximizing the preference uncertainty function $u_{P_R}(\tau_1, \tau_2)$ are used for preference feedback. In PbOP, the trajectories are then added to the trajectories buffer and the transition model is retrained for 20 (Star MDP, Gridworld) or 200 steps (Sepsis).

**Preference Feedback Collection.**   Preference labels are provided through the ground-truth reward function associated with every environment. As stated in Section 4, for computational efficiency, we sample preferences in batches of 4 (Star MDP, Gridworld) or 100 (Sepsis) to reduce the number of model updates needed.

**Policy Optimization.**   Policy optimization stages, both in estimating optimal policy sets in Sim-OPRL and PbOP and in outputting final policies, are carried out exactly through linear programming

---

**Algorithm 3** Sim-OPRL: Practical Algorithm

---

**Input:** Observational trajectories dataset $\mathcal{D}_{\text{offline}}$. Hyperparameters $\lambda_T, \lambda_R$.
**Output:** $\hat{\pi}^*$
 1: Train an ensemble $\mathcal{T}$ of transition models via bootstrapping on the observational data $\mathcal{D}_{\text{offline}}$:

$$\hat{T}(\cdot|s,a) = \frac{1}{|\mathcal{T}|} \sum_{\tilde{T} \in \mathcal{T}} \tilde{T}(\cdot|s,a); \quad u_T(s,a) = \max_{T_1, T_2 \in \mathcal{T}} |T_1(\cdot|s,a) - T_2(\cdot|s,a)|_1 \cdot R_{\max}$$

 2: $\mathcal{D}_{\text{pref}} \leftarrow \emptyset$.
 3: **for** $k = 1, ... N_p$ **do**
 4:     Estimate optimal offline policy set:

$$\Pi_{\text{offline}} = \{\pi \mid \pi = \text{argmax}_{\pi \in \Pi} \mathbb{E}_{(s,a) \sim d_{\hat{T}}^\pi} [\tilde{R}(s,a) - \lambda_T u_T(s,a)] \; \forall \tilde{R} \in \mathcal{R}\}$$

 5:     Identify exploratory policies: $\pi_1, \pi_2 = \text{argmax}_{\pi_1, \pi_2 \in \Pi_{\text{offline}}} \mathbb{E}_{\tau_1 \sim d_{\hat{T}}^{\pi_1}, \tau_2 \sim d_{\hat{T}}^{\pi_2}} [u_{P_R}(\tau_1, \tau_2)]$
 6:     Rollouts in model: $\tau_1 \sim d_{\hat{T}}^{\pi_1}, \tau_2 \sim d_{\hat{T}}^{\pi_2}$.
 7:     Collect preference label $o$ for $(\tau_1, \tau_2)$.
 8:     $\mathcal{D}_{\text{pref}} \leftarrow \mathcal{D}_{\text{pref}} \cup \{(\tau_1, \tau_2, o)\}$.
 9:     Train an ensemble $\mathcal{R}$ of reward models via bootstrapping of the preference data $\mathcal{D}_{\text{pref}}$:

$$\hat{R}(s,a) = \frac{1}{|\mathcal{R}|} \sum_{\tilde{R} \in \mathcal{R}} \tilde{R}(s,a); \quad u_R(s,a) = \max_{R_1, R_2 \in \mathcal{R}} |R_1(s,a) - R_2(s,a)|_1$$

10: **end for**
11: $\hat{\pi}^* \leftarrow \text{argmax}_{\pi \in \Pi} \mathbb{E}_{(s,a) \sim d_{\hat{T}}^\pi} [\hat{R}(s,a) - \lambda_R u_R(s,a) - \lambda_T u_T(s,a)]$

---

for the Star MDP and Gridworld using `cvxopt` (Diamond and Boyd, 2016), based on code from Lindner et al. (2021), using Proximal Policy Optimization (Schulman et al., 2017) implemented in `stable-baselines3` (Raffin et al., 2021) for the Sepsis and MiniGrid environments, and Soft Actor-Critic (Haarnoja et al., 2018) for HalfCheetah. In the latter case, after every preference collection episode, reward and policy models were trained from the checkpoint of the previous iteration, for only 20 steps to minimize computation.

**Baselines and Ablations.** We implement both OPRL baselines within our model-based offline preference-based algorithm described in Section 4. Uncertainty sampling is taking the pair with maximum preference uncertainty over 45 pairs for every sample, to reduce the load of computing preference uncertainty over the entire trajectory buffer.

Our ablation study for Figure 2 is conducted as follows. For Sim-OPRL without pessimism in the output policy, we output the policy that maximizes the value function under the MLE estimate of the transition and reward function, $\hat{T}$ and $\hat{R}$, after preference acquisition. For Sim-OPRL without pessimism in the simulated rollouts, we estimate the optimal policy set $\Pi_{\text{offline}}$ in the MLE estimate of the transition model instead of its pessimistic counterpart. Finally, for Sim-OPRL without optimism in the simulated rollouts, we generate rollouts from any two policies in $\Pi_{\text{offline}}$ instead of the most explorative ones.

## D   ENVIRONMENT DETAILS

**Star MDP.** We illustrate the transition dynamics underlying the Star MDP in Figure 5. Transition probabilities are 0.9 for all depicted solid arrows, and leave the state unchanged otherwise. Other actions also keep the state unchanged with probability 1. Episodes have length $H = 3$ and start from $s_0$. Unless specified otherwise, the offline dataset $\mathcal{D}_{\text{offline}}$ consists of 40 trajectories which only cover states $(s_0, s_1, s_3)$ and $(s_3, s_1, s_2)$.

Preferences collected over samples from the offline dataset learn slowly about the negative reward in the bottom state, as it is always included in the sampled trajectories. Instead, simulated rollouts can query a direct comparison between the optimal path and one that includes it. This example illustrates

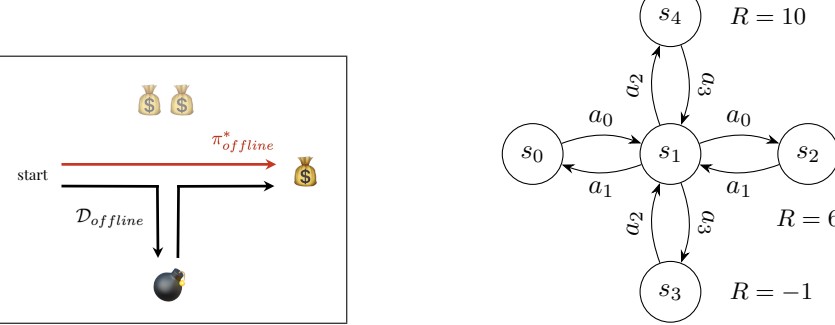

Figure 5: **Star MDP**. Transition probabilities are 0.9 for all solid arrows. Omitted actions or complementary transitions keep the state unchanged.

| Start | | $-1$ | $-1$ |
|---|---|---|---|
| | | $-1$ | $-1$ |
| | 20 | | |
| | | | 10 |

Figure 6: **Gridworld environment.** Rewards at every state are indicated if non-zero. Transition probabilities are 0.9. Thick lines indicate an obstacle, through which state transitions have probability zero.

clearly why querying feedback over simulated rollouts achieves better environment returns than over samples from the offline buffer.

**Gridworld.** We illustrate the gridworld environment in Figure 6. The environment consists of a $4 \times 4$ grid with states associated with different rewards, including a negative-reward region in the top-right corner, a high-reward but unreachable state, and a moderate-reward state at the bottom right corner. Each episode starts in the top-left corner. Transition probabilities for each of the four actions (`top`, `left`, `bottom`, `right`) are 0.9 for the intended direction, and 0.1 for the others; and action `stay` remains in the current state with probability 1. Transitions beyond the grid limits or through obstacles have probability zero, with the remainder of the probability mass for each action being distributed amongst other directions equally. The offline dataset contains 150 episodes and the behavioral policy is $\epsilon$-optimal with noise $\epsilon = 0.1$. Episodes have length $H = 10$.

**MiniGrid-FourRooms.** We also conduct experiments on the `minigrid-fourrooms-v0` D4RL dataset (Fu et al., 2020), ignoring all reward information in the offline dataset. In this environment, the agent must navigate a maze consisting of four interconnected rooms and reach a green goal square (Chevalier-Boisvert et al., 2023). Agent and goal squares are randomly placed at the beginning of every episode.

**HalfCheetah-Random.** The `halfcheetah-random-v2` dataset is also part of the D4RL benchmark (Fu et al., 2020). Our choice is motivated by Shin et al. (2022) who identify it as a particularly challenging offline preference-based reinforcement learning task. The dataset consists of 1 million transitions induced by a random policy in the MuJoCo environment Halfcheetah-v2, which rewards agents if they move forward. The observation space is 17-dimensional.

Table 4: **Transition dynamics of the sepsis simulator environment (Oberst and Sontag, 2019)**. Table adapted from (Tang and Wiens, 2021).

| Step | Variable | Current | New | Change | Variable Affected | Effect |
|---|---|---|---|---|---|---|
| 1 | Antibiotics | - | on | - | Heart Rate Systolic Blood Pressure | high → normal w.p. 0.5 high → normal w.p. 0.5 |
| | | on | off | withdrawn | Heart Rate Systolic Blood Pressure | normal → high w.p. 0.1 normal → high w.p. 0.5 |
| 2 | Ventilation | - | on | - | Oxygen Saturation | low → normal w.p. 0.7 |
| | | on | off | withdrawn | Oxygen Saturation | normal → low w.p. 0.1 |
| 3 | Vasodilators | - | on | - | Systolic Blood Pressure | low → normal w.p. 0.7 (non-diabetic) normal → high w.p. 0.7 (non-diabetic) low → normal w.p. 0.5 (diabetic) low → high w.p. 0.4 (diabetic) normal → high w.p. 0.9 (diabetic) |
| | | | | | Blood Glucose | very low, low → normal, normal → high, high → very high w.p. 0.5 (diabetic) |
| | | on | off | withdrawn | Systolic Blood Pressure | normal → low w.p. 0.1 (non-diabetic) high → normal w.p. 0.1 (non-diabetic) normal → low w.p. 0.05 (diabetic) high → normal w.p. 0.05 (diabetic) |
| 4 5 6 7 | Heart Rate Systolic Blood Pressure Oxygen Saturation Blood Glucose | | fluctuate | | | Vitals spontaneously fluctuate when not affected by treatment (either enabled or withdrawn), the level fluctuates ±1 w.p. 0.1, except: glucose fluctuates ±1 w.p. 0.3 (diabetic). |

**Sepsis Simulation.** The sepsis simulator (Oberst and Sontag, 2019) is a commonly used environment for medically-motivated RL work (Tang and Wiens, 2021). We use the original authors' publicly available code: `https://github.com/clinicalml/gumbel-max-scm/tree/sim-v2/sepsisSimDiabetes` (MIT license). The state space consists of 1,440 discrete states based on different observational variables (heart rate, blood pressure, oxygen concentration, glucose, diabetes status). The action space consists of three binary treatment options (antibiotic administration, vasopressor administration, and mechanical ventilation). The complex transition dynamics of the environment determine how each treatment affects the value of each vital sign, and are summarized in Table 4; these were designed to reflect patients' physiology (Oberst and Sontag, 2019). The ground truth reward function is sparse and only assigns a positive reward of $+1$ to surviving patients and a negative reward of $-1$ if death occurs (3 or more abnormal vitals) during their stay. The offline trajectories dataset includes 10,000 episodes following an $\epsilon$-optimal policy with noise $\epsilon = 0.1$ and the episode length is $H = 20$.

# E ADDITIONAL RESULTS

We include additional results in this section.

In Figure 7, we report the accuracy of the transition and preference model achieved for the Star MDP as we vary the size of optimality of the offline dataset. Accuracy is measured against all possible state transitions and over 100 pairs of random trajectories (random combinations of the 5 states and 4 actions in a sequence of $H = 3$). This complements our analysis in Section 7 and fig. 3. We see a steady improvement in both transition and reward model quality as we increase the amount of observational data in Figure 7a, which explains the observed dependence of $N_p$ on $N_o$ in Figure 3a.

In Figure 7b, we notice low model performance at both extremes of the x-axis. When the dataset is fully optimal, we find that all trajectories involve the same sequence of actions and states, so learning a transition or reward model from this data is challenging. We reach a similar conclusion at the other end of the spectrum at high density ratios, where the coverage the optimal states reduces. We reach highest performance for both models at intermediate values, when diversity of the observational data is high.

Still, it is important to stress that the highest accuracy of both models does not necessarily translate to the best-performing policy: good performance on the distribution induced by the optimal policy is more important, as formalized by the concentrability coefficients.

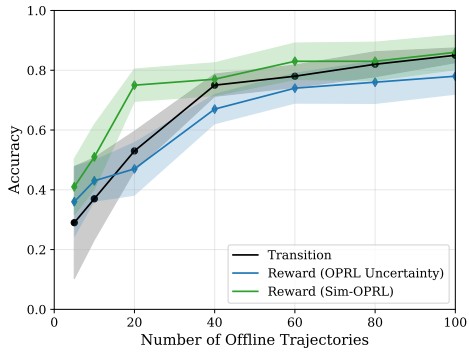
(a) As a function of offline dataset size.

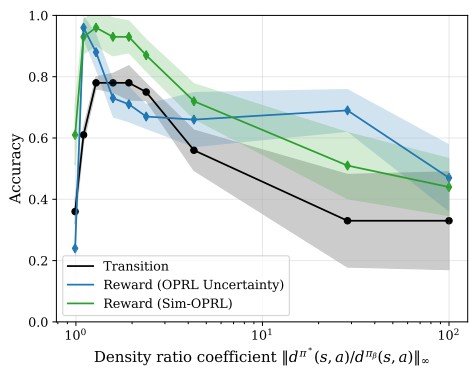
(b) As a function of optimal coverage.

Figure 7: **Transition and preference model accuracy as function of the properties of the observational data** (Star MDP). Preference elicitation is carried out until 10 preferences are queried. Mean and 95% confidence intervals over 20 experiments. Note that the transition model is the same for the two methods, as they have access to the same dataset.

Table 5: **Sample complexity $N_p$ with Sim-OPRL as a function of dataset optimality**, to reach a suboptimality gap of $\epsilon = 20$ over normalized returns for the D4RL HalfCheetah environment. Mean and 95% confidence interval over 6 experiments.

| Offline Dataset | OPRL Uniform | OPRL Uncertainty | Sim-OPRL |
| --- | --- | --- | --- |
| HalfCheetah-Random | $108 \pm 9$ | $71 \pm 8$ | $\mathbf{50 \pm 10}$ |
| HalfCheetah-Medium | $63 \pm 8$ | $\mathbf{49 \pm 6}$ | $\mathbf{36 \pm 8}$ |
| HalfCheetah-Medium-Expert | $47 \pm 8$ | $45 \pm 6$ | $\mathbf{30 \pm 8}$ |

Table 6: **Policy performance for a fixed preference budget** ($N_p = 30$) for different D4RL HalfCheetah datasets. Normalized environment returns, mean and 95% confidence interval over 6 experiments.

| Offline Dataset | OPRL Uniform | OPRL Uncertainty | Sim-OPRL |
| --- | --- | --- | --- |
| HalfCheetah-Random | $17 \pm 7$ | $\mathbf{47 \pm 10}$ | $\mathbf{49 \pm 9}$ |
| Halfcheetah-Medium | $43 \pm 6$ | $\mathbf{63 \pm 6}$ | $\mathbf{69 \pm 7}$ |
| Halfcheetah-Medium-Expert | $57 \pm 6$ | $\mathbf{72 \pm 6}$ | $\mathbf{80 \pm 6}$ |

To complement the analysis with a more complex environment, we also ran different preference elicitation algorithms on D4RL HalfCheetah datasets of varying optimality. In Table 5, we report the number of preference samples needed to achieve a target suboptimality with these different datasets. We reach the same conclusion as above: fewer preferences are needed as the dataset becomes more optimal.

In Table 6, we report policy performance for a fixed preference budget. With a fixed preference budget, policies learned from suboptimal observational datasets achieve lower returns, due to the worse quality of the transition model. Shin et al. (2022) reach similar conclusions when evaluating policy performance under different OPRL methods (in their Table 2).

## F  BROADER IMPACT

Better preference elicitation strategies for offline reinforcement learning have the potential to facilitate and improve decision-making in real-world safety-critical domains like healthcare or economics, by reducing reliance on direct environment interaction and reducing human effort in providing feedback. Potential downsides could include the amplification of biases in the offline data, potentially leading to suboptimal or unfair policies. Thorough evaluation is therefore crucial to mitigate this before deploying models in such real-world applications. In addition, human preferences may not

be fully captured by binary comparisons. As noted in our conclusion, we hope that future work will explore richer feedback mechanisms to better model complex decision-making objectives.

