# OpenReview forum: "Preference Elicitation for Offline Reinforcement Learning"
_ICLR.cc/2025/Conference — ICLR 2025 Poster_

### Official Review · Reviewer_aVUF · 2024-11-03

**Soundness:** 3
**Presentation:** 3
**Contribution:** 2
**Rating:** 8
**Confidence:** 2

**Summary:**

This paper delves into offline reinforcement learning from preference feedback and proposes an offline preference elicitation method to simulate trajectories from the learned environment model instead of sampling trajectories directly from the offline dataset. They provide theoretical justification for the previous RL with preference feedback method and show that their proposed method can effectively reduce the sample complexity upper bound. They also propose an empirical algorithm and show it can outperform prior methods and achieve SOTA on offline PbRL setups without access to the ground truth rewarded. They finally iid ablation studies to show the importance of incorporating the principle of pessimism.

**Strengths:**

1. They delve into very interesting setups: offline RL with preference feedback.

2. Their theoretical results are solid and show he advantage of their proposed preference elicitation algorithm over prior methods.

3. They propose a practical algorithm for implementation and extensive experiments show that their method outperform prior methods in several environment.

**Weaknesses:**

I do not see any big issues.

**Questions:**

/

**Details Of Ethics Concerns:**

/

---

> ### Author Response · Authors · 2024-11-14
> **Response to Reviewer aVUF**
>
> Dear Reviewer aVUF,
>
> Thank you very much for taking the time to read our work and for your very positive feedback!

---

### Official Review · Reviewer_wreC · 2024-11-04

**Soundness:** 4
**Presentation:** 4
**Contribution:** 3
**Rating:** 8
**Confidence:** 3

**Summary:**

The paper addresses the challenge of applying RL to real-world scenarios where direct interaction with the environment is impractical or unsafe. Traditional learning methods require environment interactions, which can be risky in certain fields (like healthcare applications). The paper proposes an algorithm called Sim-OPRL, an offline PBRL learning algorithm that learns from preferences without needing online interaction. This algorithm uses a learned environment model to simulate rollouts and gather preference feedback, balancing pessimism for out-of-distribution data and optimism for acquiring informative preferences. The paper formalizes the problem of preference elicitation in offline RL, proposes a novel algorithm, and provides theoretical guarantees on its sample complexity.

The paper also demonstrates the effectiveness of Sim-OPRL through empirical validation in various environments, including a gridworld and a sepsis simulation. The results show that Sim-OPRL outperforms an existing baseline algorithm (OPRL) in terms of sample efficiency and policy performance. The paper shows that by leveraging simulated rollouts, their algorithm efficiently learns the optimal policy while minimizing the number of human queries required.

**Strengths:**

1. The authors provide strong theoretical guarantees on the sample complexity of their approach, ensuring that the algorithm is both efficient and reliable. Additionally, the empirical results across various environments demonstrate the practical effectiveness and scalability of Sim-OPRL, showing it outperforms existing methods in terms of sample efficiency and policy performance.
2. Sim-OPRL incorporates a pessimistic approach to handle out-of-distribution data, ensuring robustness to model uncertainty. This is particularly important in offline settings where the data may not cover the entire state-action space. Being robust to OOD data makes the algorithm far more applicable to ‘real-world’ problems/settings.
3. The paper makes a compelling case due to their incorporation of theoretical and empirical evidence. To back up their theoretical insights, they conduct extensive experiments across two different environments. This provided empirical data confirms the practical applicability and robustness of Sim-OPRL, illustrating its effectiveness in scenarios where direct environment interaction is not feasible.
4. The attached code is well-written and largely self-documenting, with a clear and logical internal structure. This design not only facilitates ease of use for other users looking to implement the Sim-OPRL algorithm but also made the process of reviewing the practical implementation and validating the experiments straightforward and efficient. This made the review process much easier.

**Weaknesses:**

1. The paper’s empirical section does not properly consider different baseline algorithms to compare theirs with. The only algorithm that the authors use as a baseline is OPRL. This severely limits the ability to fully assess the relative performance and advantages of Sim-OPRL. To rectify this, The authors should consider including a wider array of offline PBRL algorithms/frameworks in their experiments.
2. The paper demonstrates promising results in the demonstrated environments, but it lacks validation in more complex and realistic settings. To strengthen the evidence of the algorithm’s practical applicability, the authors should evaluate Sim-OPRL on several different datasets. One example could be MuJoCo style datasets. Other relevant papers in the field construct preference datasets from the D4RL offline benchmark. These datasets provide a more challenging and ‘closer to real world’ testbed. Evaluation on such environments (in conjunction with adding more baseline algorithms) could result in a better assessment of the algorithm’s robustness, scalability, and generalizability.
3. The paper demonstrates the algorithm’s performance in relatively small-scale environments. Empirically, it does not seem to address scalability to larger, more complex environments. Due to the smaller scale test environments (Gridworld & Sepsis), the the actual scalability of the algorithm (particularly in real-world deployments outside of benchmarks) remains unclear.
4. As the authors state, for the sepsis environment, the requisite number of preference samples is rather large, due to the sparse reward function. This seems like an inherent limitation, which they posit could be solved by warm-starting the reward model. It would be interesting to see this data and how it affects performance. If a sparse reward function is a true limitation of the Sim-OPRL method, the authors should show more experiments demonstrating that this can be 'worked around' by performing warm starts. This could also help to further justify the real world applicability of the algorithm.

**Questions:**

1. How does the complexity of the reward function impact the performance of Sim-OPRL? Have you (or do you plan to) test the algorithm with environments that are characterized by more complex, multi-objective, or non-linear reward functions? If the method is agnostic to the reward function (aside from sparsity) it would help to show that as well.
2. Can you provide more details on the sensitivity of Sim-OPRL to its hyperparameters, such as the pessimism and optimism parameters? How do you recommend tuning these parameters in practice? It may be insightful to include ablation testing in the appendix that demonstrates the sensitivity (or robustness) to hyperparameter selection, especially as this could drastically affect the real-world viability of the algorithm.
3. Are there any other algorithms that would serve as a effective and informative baseline for Sim-OPRL? If not, would it be possible to run experiments that demonstrate learning performance on naive methods?
4. Could you please clarify the rationale behind limiting the experiments to the selected datasets and environments? Are there specific challenges that restrict the application of the method to a broader range of environments and dataset combinations? If there are no such constraints, additional experimental results would be valuable. Conversely, if limitations do exist, it would be beneficial to outline what they are, the reasons for their existence, and why they do not compromise the method's overall effectiveness and practical utility.
5. Generally speaking could the authors please explain the motivations for the setting further? Specifically, would it be practical to compare the results of Sim-OPRL to running standard offline RL algorithms (CQL, IQL, TD3_BC etc.) on the offline dataset directly? If not, why not?

---

> ### Author Response · Authors · 2024-11-14
> **Response to Reviewer wreC (1/2)**
>
> Dear Reviewer wreC,
>
> Thank you very much for your detailed review and feedback. We address your comments and questions below.
>
> ## Q1. Complex reward functions
>
> Other than the relationship between the reward and preference function (Equation 1), no assumptions are made on the reward function in our theoretical analysis and method development. Our method is therefore agnostic to the form of the reward function, and we validate with our experiment on the HalfCheetah environment whose reward function is non-linear (Table 2, Figure 1).
>
> As for sparsity (weakness #4), we simply propose a hypothesis that learning reward functions *from preferences* may be more challenging in general for this case, as all our baselines are inefficient in the Sepsis environment. Sim-OPRL remains much more efficient than other offline methods even in this case, which demonstrates its wider applicability.
>
> ## Q2. Hyperparameter tuning.
>
> Hyperparameters $\lambda_T , \lambda_R$ control the degree of pessimism in practice and could be
> considered equivalent to adjusting margin parameters $\beta_T , \beta_R$ in our conceptual algorithm proposed in Section 4. Since the exact values prescribed by our theoretical analysis cannot be estimated, the user must set these parameters themselves. Hyperparameter optimization in offline RL is a challenging problem in general, as the environment is not accessible to monitor policy performance, and as off-policy evaluation can be biased  (Levine et al., 2020). As a result, we fixed $\lambda_T , \lambda_R$  for both our method and all baselines to avoid giving an unfair advantage to one or the other.
>
> Our work focuses on sample efficiency, an important bottleneck in real-world applications where interaction with domain experts can be costly or time-consuming. Improving preference modeling and policy optimization falls outside the scope of our problem setting, presented in Section 3.
>
> ## Q3. Additional baselines
>
> We would like to clarify that our work is not concerned with preference modeling or policy optimization, but with efficient **preference elicitation** in the offline context. In Section 2 and Appendix B of our manuscript, we discuss related works contributing to the former research direction (see references below), but we note that their *elicitation strategy* is either based on a fixed preference dataset (equivalent to OPRL uniform sampling) or on uncertainty-sampling (equivalent to OPRL uncertainty sampling).
>
> For instance, Brown et al., (2019, 2020); Park et al. (2022); Kim et al. (2023); Hejna and Sadigh, (2024) all assume access to a fixed preference dataset. PEBBLE (Lee et al., 2021) performs uncertainty sampling and also assumes access to the environment.
>
> We benchmark our elicitation strategy with these two methods. To the best of our knowledge, however, there are no additional baselines for preference elicitation from offline data. Naturally, if you are aware of alternative offline elicitation methods, we would be happy to run these. We hope this answer also addresses your weakness #1.
>
> ## Q4. Environment complexity and scalability
>
> We are surprised by your comments and questions on our experimental setup, as we report empirical performance on a range of environments in Table 2 and Figure 1. For instance, the MuJoCo HalfCheetah and Minigrid environments are from the D4RL benchmark (Fu et al., 2020), which are identified as particularly challenging offline preference-based reinforcement learning tasks (Shin et al., 2022). As detailed in Appendix D, our environments consist of high-dimensional state spaces with continuous or discrete action spaces, follow complex transition dynamics, and have sparse rewards and termination conditions. This makes them representative of the challenge of learning a reward function and learning offline in a real-world, large-scale application. We hope this addresses your comments regarding weaknesses #2 and #3.

---

> > ### Author Response · Authors · 2024-11-14
> > **Response to Reviewer wreC (2/2)**
> >
> > ## Q5. Offline RL baselines
> >
> > Our problem setting assumes we have no reward signal in our observational dataset. In real-world data, it is unlikely that every observed state-action pair would be annotated with a numerical reward signal. As a result, offline RL methods are not applicable to our setting. Instead, we assume we can ask experts for their preferences over trajectories, and we ask how to collect their feedback as efficiently as possible.
> >
> > As an upper bound, or target, for the performance of our preference elicitation method, we report the performance of policy trained with the ground truth reward function. This corresponds to the optimal offline policy ($\pi^*_{\textrm{offline}}$) in Figure 1, obtained with the equivalent of MOPO (Yu et al., 2020) as we consider a model-based setting.
> >
> > We stress again that our work is concerned with improving efficiency of preference elicitation in the offline setting. Algorithms like CQL, IQL, TD3_BC are not preference-based methods and are concerned with the modeling or policy optimization process, which is orthogonal to this work.
> >
> > ---
> >
> > Thank you again for your review. We look forward to hearing your thoughts on our clarifications. We hope to have addressed your concerns and that you will increase your score, in light of your otherwise positive feedback.
> >
> >
> > ### References
> >
> > D. Brown, R. Coleman, R. Srinivasan, and S. Niekum. Safe imitation learning via fast Bayesian reward inference from preferences. International Conference on Machine Learning, 2020.
> >
> > D. Brown, W. Goo, P. Nagarajan and S. Niekum. Extrapolating beyond suboptimal demonstrations via inverse reinforcement learning from observations. International conference on machine learning. 2019.
> >
> > J. Fu, A. Kumar, O. Nachum, G. Tucker, and S. Levine. D4rl: Datasets for deep data-driven reinforcement learning. arXiv preprint arXiv:2004.07219, 2020.
> >
> > J. Hejna and D. Sadigh. Inverse preference learning: Preference-based rl without a reward function. Advances in Neural Information Processing Systems, 36, 2024.
> >
> > C. Kim, J. Park, J. Shin, H. Lee, P. Abbeel, and K. Lee. Preference transformer: Modeling human preferences using transformers for rl. arXiv preprint arXiv:2303.00957, 2023.
> >
> > K. Lee, L. Smith, and P. Abbeel. Pebble: Feedback-efficient interactive reinforcement learning via relabeling experience and unsupervised pre-training. arXiv preprint arXiv:2106.05091, 2021.
> >
> > S. Levine, A. Kumar, G. Tucker, and J. Fu. Offline reinforcement learning: Tutorial, review, and perspectives on open problems. arXiv preprint arXiv:2005.01643, 2020.
> >
> > J. Park, Y. Seo, J. Shin, H. Lee, P. Abbeel, and K. Lee. Surf: Semi-supervised reward learning with data augmentation for feedback-efficient preference-based reinforcement learning. arXiv preprint arXiv:2203.10050, 2022.
> >
> > D. Shin, A. Dragan, and D. S. Brown. Benchmarks and algorithms for offline preference-based reward learning. Transactions on Machine Learning Research, 2022.
> >
> > T. Yu, G. Thomas, L. Yu, S. Ermon, J. Y. Zou, S. Levine, C. Finn, and T. Ma. Mopo: Model-based offline policy optimization. Advances in Neural Information Processing Systems, 33:14129–14142, 2020.

---

> ### Author Response · Authors · 2024-11-24
>
> Dear Reviewer wreC,
>
> Thank you again for your review. We were wondering if you had had the opportunity to read our rebuttal, as we look forward to discussing your follow-up thoughts before the end of the discussion period. We hope you will consider increasing your score if your concerns have been addressed.

---

> > ### Author Response · Authors · 2024-11-29
> >
> > Dear Reviewer wreC,
> >
> > Thank you very much for increasing your score. We are glad your concerns have been addressed.

---

### Official Review · Reviewer_BKXk · 2024-11-04

**Soundness:** 3
**Presentation:** 3
**Contribution:** 3
**Rating:** 6
**Confidence:** 4

**Summary:**

The paper presents Sim-OPRL, an offline preference-based reinforcement learning algorithm that addresses the challenge of acquiring preference feedback in a fully offline setup. It leverages a learned environment model to elicit preference feedback on simulated rollouts, balancing conservatism and exploration. The main idea is to employ a pessimistic estimation for the transition dynamics (based on the offline dataset) for the OOD issue, and use an optimistic estimation for the reward model (based on the preference elicitation data). The benefit of using simulated rollouts is to avoid wasting preference budget on trajectories with low rewards. The authors provide theoretical guarantees on sample complexity and demonstrate the empirical performance of a practical version of Sim-OPRL across various environments, showing its superiority over previous baseline methods (OPRL and PbOP).

**Strengths:**

- This paper focuses on the preference elicitation problem on offline RL, which attracts wide attention recently from many fields (such as RLHF for LLMs).
- This paper has theoretical results on the proposed algorithm with some high-level insights (e.g., pessimism for dynamics and optimism for reward modeling).
- This paper has practical algorithm designs and good empirical results.

**Weaknesses:**

- **Complexity of Implementation:** The algorithm's reliance on learning several accurate dynamics model might be challenging in practice, especially if the model fails to capture the true dynamics. Moreover, Sim-OPRL requires the trajectory rollouts using the dynamics model and the error may accumulate, which poses higher requirements for the dynamics model. Do the authors have any idea on how to design practical algorithms with less computational overhead (e.g., estimating multiple models) and on more complex environments (e.g., when it is hard to learn an accurate dynamics model).
- **Lack of study on the dependence on quality of offline data and feedback:** The performance of Sim-OPRL may be heavily dependent on the quality and coverage of the offline dataset. For the experiments in on the tasks listed in Table 2, how are the offline datasets are collected? Are they expert datasets (so the concentrability coefficients are small)? How the feedback is generated in the experiments? How would the algorithm perform when we vary the feedback quality?
- Minor: What is ``\hat{R}_\text{inf}``? I can guess it is pessimistic reward, but ``\hat{R}_\text{inf}`` and ``\hat{T}_\text{inf}`` are not formally defined.

**Questions:**

- I do not quite understand “An advantage of sampling from the offline buffer, however, is that it is not sensitive to the quality of the model” in L346. What does “the model” refer to?
- Should $N_T$ in the second equation in L369 be $N_R$?

---

> ### Author Response · Authors · 2024-11-14
> **Response to Reviewer BKXk (1/2)**
>
> Dear Reviewer BKXk,
>
> Thank you very much for your positive feedback and for your review. We address your comments below.
>
> ## Q1. Complexity of implementation.
>
> While Sim-OPRL requires learning a transition model from the observational data for simulating rollouts, we note that this modeling step (and the ensemble needed for uncertainty estimation) is often also needed for model-based offline RL algorithms such as MOPO (Yu et al., 2020). To address your concern that our algorithm might not perform as well on complex environments, we find that Sim-OPRL also outperforms baselines on the complex D4RL datasets and in the Sepsis simulation (Table 2 and Figure 1).
>
> We agree that modeling complexity increases as we go from uniform-sampling OPRL, to uncertainty-sampling OPRL (which now requires uncertainty estimation for the reward function), and finally to Sim-OPRL  (which requires uncertainty estimation for both reward and transition functions). Considering our results in Table 2, we see that greater complexity leads to higher performance. For a method with “less computational overhead”, practitioners can therefore use one of our baselines, at the cost of performance.
>
>
> ## Q2. Dependence on dataset optimality and feedback quality.
>
> **Dataset Optimality.** We explore datasets with different levels of policy optimality, ranging from random policy (e.g., Halfcheetah-Random) to $\epsilon$-optimal (e.g., Sepsis). Further details on each dataset and environment are provided in Appendix D. We conclude that Sim-OPRL is more efficient than OPRL baselines in all cases.
>
> Following Theorems 5.1 and 6.1, if the behavioral policy covers the optimal policy well, the concentrability terms will be smaller: fewer preferences are needed to achieve a target suboptimality. The reverse is true when the behavioral policy is far from optimal. We also discuss this in detail in our response to Reviewer tF7g. For a rigorous empirical analysis of performance as a function of dataset optimality, we propose an ablation in Figure 3b. We measure optimality through the density ratio coefficient which upper bounds $C_T$. Our empirical results are discussed in the paragraph starting line 512, and support the above theoretical analysis.
>
> To complement this with a more complex environment, we also ran Sim-OPRL on Halfcheetah medium and medium-expert datasets. In the following table, we report the sample complexity achieved with these different datasets to reach a suboptimality gap of $\epsilon = 20$ over normalized returns. We reach the same conclusion as above: fewer preferences are needed as the dataset becomes more optimal. We will include this additional result in our revised manuscript.
>
> | Offline Dataset |    Sample Complexity $N_p$    |
> |---------------------------|:-----------:|
> | halfcheetah-random        | 50 $\pm$ 10 |
> | halfcheetah-medium        |  36 $\pm$ 8 |
> | halfcheetah-medium-expert |  30 $\pm$ 8 |
>
> **Feedback Quality.** Following prior empirical and theoretical work on preference elicitation for reinforcement learning (Chen et al., 2022, Zhan et al.,2023a and 2023b, Shin et al., 2022), our framework assumes the existence of a ground-truth reward function $R$, and that the feedback we receive follows the Bradley-Terry model determined by $R$. We formulate this problem in Section 3.
>
> We therefore use each environment’s true reward function to generate preference feedback whenever our algorithms query it, again following prior work. Introducing noise in the feedback model violates this assumption, and therefore falls outside of the scope of our analysis.
>
> ## Q3. Additional questions
> Thank you for spotting these unclear elements and typos.
> - The pessimistic transition and reward models are defined as follows: $\hat{R}\_{inf},\hat{T}\_{inf} = argmin_{\tilde{R} \in \mathcal{R}, \tilde{T} \in \mathcal{T}} \max_{\pi \in \Pi} V^{\pi}_{\tilde{T}, \tilde{R}}$. We will include this definition in our revised manuscript.
> - The model refers to the transition model. We propose to rewrite the sentence for clarity: “While sampling from the offline buffer in OPRL is not sensitive to the quality of the transition model, good coverage of the optimal policy is needed from both transition and preference data to achieve low suboptimality.”
> - Thank you very much for spotting this typo.
>
> ---
> Thank you again for your positive feedback and interesting comments. We hope our response addresses your remaining concerns, and we would be very grateful if you would increase your score.

---

> > ### Author Response · Authors · 2024-11-14
> > **Response to Reviewer BKXk (2/2)**
> >
> > ### References
> >
> > X. Chen, H. Zhong, Z. Yang, Z. Wang, and L. Wang. Human-in-the-loop: Provably efficient preference-based reinforcement learning with general function approximation. In International Conference on Machine Learning, 2022.
> >
> > J. Fu, A. Kumar, O. Nachum, G. Tucker, and S. Levine. D4rl: Datasets for deep data-driven reinforcement learning. arXiv preprint arXiv:2004.07219, 2020.
> >
> > D. Shin, A. Dragan, and D. S. Brown. Benchmarks and algorithms for offline preference-based reward learning. Transactions on Machine Learning Research, 2022.
> >
> > T. Yu, G. Thomas, L. Yu, S. Ermon, J. Y. Zou, S. Levine, C. Finn, and T. Ma. Mopo: Model-based offline policy optimization. Advances in Neural Information Processing Systems, 33:14129–14142, 2020.
> >
> > W. Zhan, M. Uehara, N. Kallus, J. D. Lee, and W. Sun. Provable offline reinforcement learning with human feedback. In ICML 2023 Workshop The Many Facets of Preference-Based Learning, 2023a.
> >
> > W. Zhan, M. Uehara, W. Sun, and J. D. Lee. How to query human feedback efficiently in rl? In ICML 2023 Workshop The Many Facets of Preference-Based Learning, 2023b.

---

> > > ### Author Response · Authors · 2024-11-24
> > >
> > > Dear Reviewer BKXk,
> > >
> > > Thank you again for your review.
> > >
> > > We have now updated our manuscript with additional experiments varying the optimality of the observational dataset. Tables 5 and 6 now include performance for different datasets and preference elicitation methods, confirming the conclusions drawn above and the superiority of Sim-OPRL.
> > >
> > > We look forward to discussing your follow-up thoughts and hope you will consider increasing your score if your concerns have been addressed.

---

> > > > ### Author Response · Authors · 2024-11-29
> > > >
> > > > Dear Reviewer BKXk,
> > > >
> > > > With the extended discussion period, we hope to have the opportunity to discuss your thoughts on our rebuttal. Your continued support is important to us, and we want to ensure your concerns are addressed.

---

### Official Review · Reviewer_8W5G · 2024-11-04

**Soundness:** 3
**Presentation:** 3
**Contribution:** 3
**Rating:** 6
**Confidence:** 3

**Summary:**

This paper uses the offline dataset to learn the environment model. They do not assume they have access to the reward in the offline data set. Such offline datasets contribute to the overall learning by providing an estimation of the transition probability. This paper provides a theoretical analysis of reinforcement learning with offline datasets to achieve preference elicitation. The experiments show their algorithms outperform other algorithms in several environments. They also conducted an ablation test to show the importance of pessimistic with respect to the transition model.

**Strengths:**

Strengths:
1. This paper provides a good theoretical analysis of preference elicitation with the offline datasets. It bounds the value difference between the optimal policy under the estimated transition model and the true optimal policy. Such bounds are achieved by decomposing the loss from the model estimation and the reward estimation.
2. Experiments show the proposed methods outperform other algorithms in several environments.
3. This paper conducted an ablation study to show the importance of pessimistic with respect to the transition model.

**Weaknesses:**

Weaknesses:

1. The experiment environments are relatively simple. The grid world is quite small. It is interesting to try to extend this to more challenging reinforcement learning benchmarks.

**Questions:**

N/A

---

> ### Author Response · Authors · 2024-11-14
> **Response to Reviewer 8W5G**
>
> Dear Reviewer 8W5G,
>
> Thank you very much for your positive feedback and for your review.
>
> We are surprised by your comment on our experimental setup, as we report empirical performance on a range of environments in Table 2 and Figure 1. Among others, we explore environments from the D4RL benchmark (Fu et al., 2020) identified as particularly challenging offline preference-based reinforcement learning tasks (Shin et al., 2022), as well as a medical simulation designed to model the evolution of patients with sepsis (Oberst and Sontag, 2019). As detailed in Appendix D, these environments consist of high-dimensional state spaces with continuous or discrete action spaces, follow complex transition dynamics, and have sparse rewards and termination conditions. This makes them representative of the challenge of learning a reward function and learning offline in a real-world application.
>
> ---
> Thank you again for your positive feedback. We look forward to hearing your thoughts in follow-up. We hope our response addresses your remaining concern and we would be very grateful if you would increase your score.
>
> ### References
>
> J. Fu, A. Kumar, O. Nachum, G. Tucker, and S. Levine. D4rl: Datasets for deep data-driven reinforcement learning. arXiv preprint arXiv:2004.07219, 2020.
>
> M. Oberst and D. Sontag. Counterfactual off-policy evaluation with gumbel-max structural causal models. In International Conference on Machine Learning, pages 4881–4890. PMLR, 2019.
>
> D. Shin, A. Dragan, and D. S. Brown. Benchmarks and algorithms for offline preference-based reward learning. Transactions on Machine Learning Research, 2022.

---

> > ### Author Response · Authors · 2024-11-24
> >
> > Dear Reviewer 8W5G,
> >
> > Thank you again for your review. We were wondering if you had had the opportunity to read our response, as we believe it may address your concern regarding our experimental setup. We look forward to discussing your thoughts and hope you will consider increasing your score.

---

> > > ### Author Response · Authors · 2024-11-29
> > >
> > > Dear Reviewer 8W5G,
> > >
> > > With the extended discussion period, we hope to have the opportunity to discuss your thoughts on our rebuttal. Your continued support is important to us, and we want to ensure your concerns are addressed.

---

### Official Review · Reviewer_tF7g · 2024-11-06

**Soundness:** 3
**Presentation:** 3
**Contribution:** 2
**Rating:** 6
**Confidence:** 2

**Summary:**

This paper studies preference-based reinforcement learning (PbRL) in offline setting, in which the agent utilizes a fixed trajectory dataset for policy learning and can query humans for preference feedback. In particular, the authors propose to sample preference queries by rolling out trajectory data using learned models of MDPs. The authors provides theoretical guarantees for the sample complexity of their proposed strategy and verify it on simple control tasks.

**Strengths:**

The idea of using simulated rollouts in preference queries is a natural but unexplored idea in the literature of PbRL. One strength of this paper is that, the authors show the effectiveness in terms of sample complexity both theoretically and empirically.

**Weaknesses:**

My concern is about the quality of learned policies. While I agree with the optimality criterion mentioned in 3.2, I think to ensure the practical value of the proposed strategy, it is important to include evaluations for offline dataset of varying optimality. This is because for high-dimensional tasks, under a fixed budget of offline trajectories, the coverage over state-action space and the optimality of the behavior policy, can be conflicting objectives. The state-action space is less covered by good behavior policies, yet this reduced coverage can raise concerns on learned transition model. See detailed question below.

**Questions:**

1. Based on your theoretical analysis, could you discuss how you expect the performance will change on dataset of varying optimality?
2. Could you present experiment results on other dataset for the Cheetah environment, such as medium, medium-expert and expert, to support your discussion?

---

> ### Author Response · Authors · 2024-11-14
> **Response to Reviewer tF7g**
>
> Dear Reviewer tF7g,
>
> Thank you very much for your positive feedback and for taking the time to review our paper.
>
> We agree that the optimality of the behavior policy has an important effect on performance. We summarize our analysis and propose additional results in the following paragraphs.
>
> ## Q1. Theoretical analysis
> Your intuition about optimality affecting the quality of the transition model is right. For Sim-OPRL, as you noted, the transition model mostly needs to be accurate in the state-action space corresponding to the optimal policy, since that is where we generate rollouts and optimize the final policy.
>
> Therefore, if the behavioral policy does not cover the optimal policy, the transition model will be less accurate in this region. Following our theoretical analysis, the concentrability term $C_T$ will be very large. This means the transition term in Theorems 6.1 will be large. More preferences $N_p$ will be needed to achieve the same suboptimality.
>
> Conversely, if the behavioral policy covers the optimal policy well, the concentrability terms will be smaller. Fewer preferences would be needed to achieve a target suboptimality.
>
> The same conclusions apply to SoTA offline preference elicitation methods (OPRL, Shin et al., 2022), as we find in Theorem 5.1 that their suboptimality depends on concentrability terms $C_T$ and $C_R$ for both transition and reward terms. Under a very suboptimal behavior policy, many preferences are needed to overcome the concentrability terms dominating the transition and reward terms respectively. As the behavior policy becomes closer to the optimal policy, we may recover good performance with fewer preferences.
>
> ## Q2. Empirical analysis
>
> Figure 3b in our manuscript proposes an ablation measuring sample efficiency as a function of the optimality of the behavior policy. We measure optimality through the density ratio coefficient which upper bounds $C_T$. Our empirical results are discussed in the paragraph starting line 512, and support the above theoretical analysis.
>
> We also ran Sim-OPRL on Halfcheetah medium and medium-expert datasets. In the following table, we report the number of preferences needed to achieve a suboptimality gap of $\epsilon = 20$ over normalized returns. We reach the same conclusion on this environment: as the dataset becomes more optimal, fewer preferences are needed to achieve a given suboptimality.
>
> | Offline Dataset |    Sample Complexity $N_p$    |
> |---------------------------|:-----------:|
> | halfcheetah-random        | 50 $\pm$ 10 |
> | halfcheetah-medium        |  36 $\pm$ 8 |
> | halfcheetah-medium-expert |  30 $\pm$ 8 |
>
>  We will include this additional result in our revised manuscript.
>
>
> ---
> Thank you again for your positive feedback and insightful questions. We hope our rebuttal addresses your remaining concerns, and we would be very grateful if you would increase your score.
>
> ### References
>
> D. Shin, A. Dragan, and D. S. Brown. Benchmarks and algorithms for offline preference-based reward learning. Transactions on Machine Learning Research, 2022.

---

> > ### Comment · Reviewer_tF7g · 2024-11-20
> >
> > Thanks for your reply. As mentioned in the strength section, I agree that your method has advantages in terms of sample complexity. But as I mentioned in the Weakness part, I am concerning the performance of the policy, which is also an important factor in practice. Your additional experiments are still only about sample complexity.

---

> > > ### Author Response · Authors · 2024-11-21
> > >
> > > Dear Reviewer tF7g,
> > >
> > > Thank you for clarifying your concern. We agree that policy performance is a critical evaluation metric, and that is why our sample complexity analysis measures how many preferences are needed to achieve close-to-optimal performance (within 20% of optimal returns).
> > >
> > > In the following, we report policy performance (normalized environment returns), for a fixed preference budget ($N_p=30$).
> > >
> > > | Environment               | OPRL Uniform | OPRL Uncertainty |  Sim-OPRL  |
> > > |---------------------------|--------------|------------------|:----------:|
> > > | halfcheetah-random        | 17 $\pm$ 8   | 47 $\pm$ 10      | 49 $\pm$ 9 |
> > > | halfcheetah-medium        | 43 $\pm$ 6   | 63 $\pm$ 6       | 69 $\pm$ 7 |
> > > | halfcheetah-medium-expert | 57 $\pm$ 6   | 72 $\pm$ 6       | 80 $\pm$ 6 |
> > >
> > > These results support our above analysis and confirm your expectation that, with a fixed preference budget, policies learned from suboptimal observational datasets achieve lower returns, due to the worse quality of the transition model. Shin et al., 2022, reach similar conclusions when evaluating policy performance under different OPRL methods (in their Table 2). Sim-OPRL performs comparably or better than offline preference elicitation baselines.
> > >
> > > We have updated our manuscript with this table in Appendix E (page 28). This analysis complements Figure 1 in our manuscript, which plots the performance of the policy as a function of the number of preferences collected, for all of the other datasets and environments considered.
> > >
> > > We hope our answer addresses your concern and look forward to hearing your thoughts.

---

> > > > ### Author Response · Authors · 2024-11-25
> > > >
> > > > Dear Reviewer tF7g,
> > > >
> > > > As the discussion phase will be ending soon, we look forward to hearing your thoughts on our latest results. We hope you will consider increasing your score if your concerns have been addressed.

---

> > > > > ### Author Response · Authors · 2024-11-29
> > > > >
> > > > > Dear Reviewer tF7g,
> > > > >
> > > > > With the extended discussion period, we hope to have the opportunity to discuss your thoughts on our rebuttal. Your continued support is important to us, and we want to ensure your concerns are addressed.

---

### Author Response · Authors · 2024-11-14
**Response to All Reviewers**

Dear Reviewers,

Many thanks for taking the time to review our work.

We are grateful for your positive feedback, noting the challenge of the problem considered and its importance to the wider community. Our proposed algorithm was described as a “natural but unexplored idea” with a “compelling case”. Reviewers praised the combination of **insightful theoretical validation** and **strong empirical results**, with a practical implementation and extensive experiments.

We address reviewers’ questions individually and will incorporate their feedback in our manuscript. We summarize the main discussion points below.
- For Reviewers 8W5G and wreC, we clarify our experimental setup. Our paper considers a range of environments, including **complex, high-dimensional environments from established RL benchmarks**. This demonstrates the scalability and general applicability of our practical algorithm.
- For Reviewers tF7g and BKXk, we detail our **ablation study on the optimality of the behavior policy**. Both theory and experiments suggest that more preferences are needed when the observational dataset is suboptimal.
- For Reviewer BKXk, we explain that the only existing baselines for the problem of offline preference **elicitation** are sampling trajectories (1) randomly or (2) through uncertainty-sampling from the observational dataset. Offline RL and PbRL methods do not propose alternative elicitation strategies.

Thank you again for your valuable suggestions, we look forward to hearing your thoughts. We hope you will consider increasing your scores if we have addressed your remaining concerns.

---

### Author Response · Authors · 2024-11-18
**Manuscript Revision**

Dear Reviewers,

Thank you again for your reviews. We have updated our manuscript based on your feedback and questions, with changes highlighted in blue.

The main changes concern Section 6.2, where we complete our theoretical analysis of sample complexity as a function of dataset optimality. Our additional results on the HalfCheetah environments are included in Appendix E and mentioned in the main body (line 521).

We look forward to hearing your follow-up thoughts on our rebuttal.

---

### Meta-Review · Area_Chair_zUDF · 2024-12-23

**Metareview:**

This paper studies preference-based reinforcement learning (PbRL) in offline setting, a relatively unexplored area.
The authors show the sample complexity effectiveness of their approach.
The main contribution of the paper is theoretical; however, they derive a practical algorithm that works well in practice.
Perhaps the major weakness is that empirical experiments concerns environments that are quite simple.

**Additional Comments On Reviewer Discussion:**

None

---

### Decision · Program_Chairs · 2025-01-22

Accept (Poster)